# ArrayDPS: Unsupervised Blind Speech Separation with a Diffusion Prior

**Zhongweiyang Xu** [1]  **Xulin Fan** [1]  **Zhong-Qiu Wang** [2]  **Xilin Jiang** [3]  **Romit Roy Choudhury** [1]

## Abstract

Blind Speech Separation (BSS) aims to separate multiple speech sources from audio mixtures recorded by a microphone array. The problem is challenging because it is a blind inverse problem, i.e., the microphone array geometry, the room impulse response (RIR), and the speech sources, are all unknown. We propose **ArrayDPS** to solve the BSS problem in an unsupervised, array-agnostic, and generative manner. The core idea builds on diffusion posterior sampling (DPS), but unlike DPS where the likelihood is tractable, ArrayDPS must approximate the likelihood by formulating a separate optimization problem. The solution to the optimization approximates room acoustics and the relative transfer functions between microphones. These approximations, along with the diffusion priors, iterate through the ArrayDPS sampling process and ultimately yield separated voice sources. We only need a simple single-speaker speech diffusion model as a prior, along with the mixtures recorded at the microphones; no microphone array information is necessary. Evaluation results show that ArrayDPS outperforms all baseline unsupervised methods while being comparable to supervised methods in terms of SDR. Audio demos and codes are provided at: https://arraydps.github.io/ArrayDPSDemo/ and https://github.com/ArrayDPS/ArrayDPS.

## 1. Introduction

The cocktail party problem is a classic challenge in audio signal processing and machine learning (Cherry, 1953; McDermott, 2009). It arises when multiple speakers talk simultaneously in the same room, and several microphones capture their voices. Each microphone records a mixture of all the speakers' voices. The objective is to separate these mixtures to recover the individual voice sources. In recent years, supervised learning based methods have shown remarkable potential to solve the cocktail party problem (Wang & Chen, 2018). However, these methods are usually trained with supervision from speech datasets that were synthesized by using acoustic simulators. Such synthetic supervision inherits several problems:

**(1) Generalizability:** The simulated dataset does not match real-world acoustic environments, causing model generalization issues (Pandey & Wang, 2020; Zhang et al., 2021).

**(2) Deterministic:** These models are trained to be deterministic, and hence they output one fixed separation solution for a given mixture. This can give blurred results when the solution is not unique (Jayaram & Thickstun, 2020), i.e., the probability distribution of the sources, given the mixture, is a multi-modal distribution.

**(3) Fixed Array Geometry:** These models usually assume the geometry of microphone arrays is fixed and known, preventing flexibility to unseen arrays (Taherian et al., 2022).

To address all the problems above, we propose ArrayDPS, a generative, unsupervised, and array-agnostic algorithm for speech separation, which fully exploits speech prior information. Building on the diffusion posterior sampling (DPS) technique (Chung et al., 2023b; Song et al., 2023), ArrayDPS treats speech separation as an inverse problem. Briefly, our goal is to recover speech sources $s$ from multi-microphone mixture measurements $x = A(s) + \epsilon$, where $A(\cdot)$ denotes the source mixing process in reverberant conditions. DPS samples from $p(s|x)$ by using a pre-trained diffusion prior $p(s)$, and a tractable likelihood model $p(x|s)$. While we use a pre-trained speech source diffusion prior as well, unfortunately, the likelihood is intractable in our case. This is because the distortion function $A(\cdot)$ depends not only on the unknown array geometry but also on the unknown RIRs over which the speech arrives at each microphone. Without any knowledge of the array geometry, the RIRs, and the speech sources, this is referred to as a "blind" separation problem.

To solve the problems mentioned above, at each diffusion sampling step, with the current source estimate $\hat{s}$, we estimate $A$ by: $\hat{A} = \arg\max_{A} p(x|\hat{s}, A)$. Then we use $p(x|\hat{s}, \hat{A})$

[1]Department of Electrical and Computer Engineering, University of Illinois Urbana-Champaign, Champaign, USA [2]Department of Computer Science and Engineering, Southern University of Science and Technology, Shenzhen, China [3]Columbia University, NYC, USA. Correspondence to: Romit Roy Choudhury <croy@illinois.edu>, Zhongweiyang Xu <zx21@illinois.edu>.

*Proceedings of the $42^{st}$ International Conference on Machine Learning*, Vancouver, Canada. PMLR 267, 2025. Copyright 2025 by the author(s).

as a tractable approximation for the intractable likelihood $p(x|\hat{s})$. Lastly, similar to DPS, we can use the prior score and our approximated likelihood score to get the posterior score, which allows posterior sampling for separation.

We borrow ideas from Forward Convolutive Prediction (Wang et al., 2021a;b; Wang & Watanabe, 2023) for the estimation of $A$ mentioned above, and we use Independent Vector Analysis (IVA) (Kim et al., 2006; Hiroe, 2006) for initialization to improve sampling stability.

Extensive evaluation shows that ArrayDPS can achieve similar performance against recent supervised methods evaluated on ad-hoc microphone arrays, and performs the best among all unsupervised blind speech separation algorithms. The gains arise from using stronger speech priors from the diffusion models, as opposed to other unsupervised methods (Wang & Watanabe, 2023; Tran Vu & Haeb-Umbach, 2010; Kim et al., 2006) that do not fully exploit speech source priors. As a result, they often suffer from problems like frequency permutation or spatial aliasing, which prevents correct separation. In contrast, ArrayDPS leverages the diffusion prior which automatically bypasses those problems.

Our main contributions are summarized as follows:
**Unsupervised:** Only a clean-speech pre-trained diffusion model is needed, mitigating the generalization issues that affect supervised methods.
**Array-Agnostic:** ArrayDPS does not rely on array patterns and can generalize on any microphone array geometries.
**Generative:** The method samples from the posterior, allowing multiple plausible separation results while fully exploiting the speech source prior.
**DPS for Multi-channel:** ArrayDPS is the first method to solve the multi-channel array inverse problem with DPS; we believe that it can enable many other applications beyond speech separation in multi-channel array signal processing.

## 2. Problem Formulation

In a reverberant environment, assume a $C$-channel microphone array is recording $K$ concurrent speakers. Let us denote the $K$ clean speech sources as $\tilde{s}_1(t), \tilde{s}_2(t), ..., \tilde{s}_K(t) \in \mathbb{R}$, where $t \in \{0, 1, 2, ..., T-1\}$ is the sample index of the waveform. These clean speech sources get filtered by the room impulse responses (RIR) and arrive at the $C$ microphones, forming the reverberant speech source images $s_{k,c}(t) \in \mathbb{R}$, where $k \in \{1, 2, ..., K\}$ indexes the sources and $c \in \{1, 2, ..., C\}$ indexes the microphones. Thus, each microphone captures a mixture of $K$ source images and some measurement noise, and we denote this mixture as $x_c(t) \in \mathbb{R}$. The speaker-to-microphone filtering and mixing process can be modeled as:

$$s_{k,c}(t) = h_{k,c}(t) * \tilde{s}_k(t) \qquad (1)$$

$$x_c(t) = \sum_{k=1}^{K} s_{k,c}(t) + n_c(t), \qquad n_c(t) \sim \mathcal{N}(0, \sigma_n^2 I) \quad (2)$$

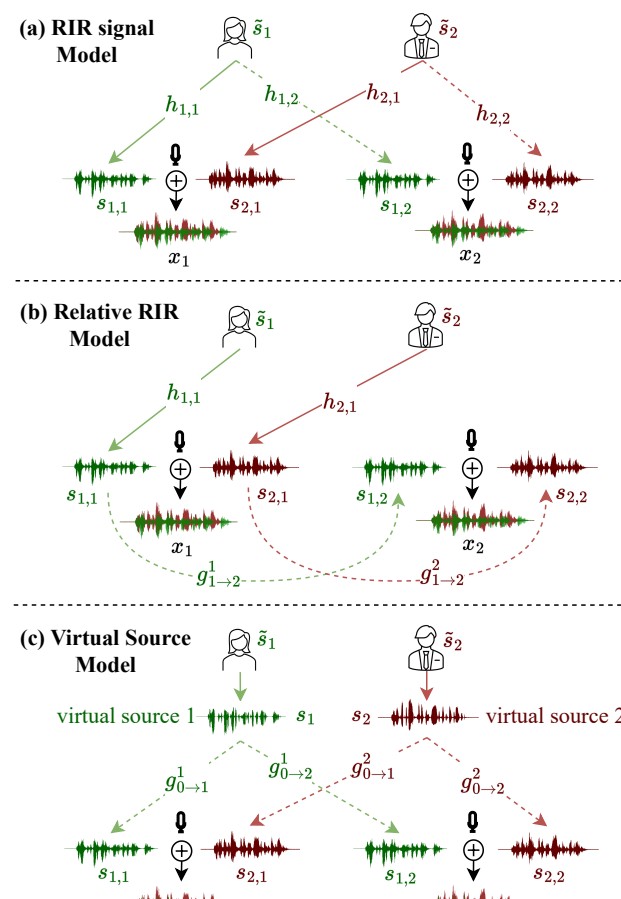

*Figure 1.* (a) Signal mixing in the real world; (b) Relative RIR model; and (c) Relative RIR from virtual sources to real channels. Measurement noise is ignored in the figures.

where $h_{k,c}(t)$ is the RIR from the $k^{th}$ source location to the $c^{th}$ microphone, and $*$ is the convolution operation. $n_c(t)$ is the $c^{th}$ microphone's white noise. Fig. 1(a) illustrates the signal model for the case of $C = 2$ and $K = 2$. **Our goal is to extract all $K$ reverberant speech source images at the reference channel** ($c = 1$), **i.e., extract $s_{1:K,1}$ from all the mixtures $x_{1:C}$, without assuming any microphone array geometry, any source location, or any supervision**[1].

We approach this problem through a generative model, i.e., sampling from $p(s_{1:K,1}|x_{1:C})$. Before formulating that, we first discuss the relative room impulse response (relative RIR) model often useful for microphone arrays (shown in Fig. 1(b)). The relative RIR models the linear relationship between any given channel and a designated reference channel (conventionally denoted as $c_1 = 1$). Note that from Eq. 1, there exists a linear filter that can filter $s_{k,c_1}(t)$ to

---

[1]For convenience, we use $s_{k_1:k_2,c_1:c_2}$ to denote $\{s_{k,c}(t)|c_1 \le c \le c_2, k_1 \le k \le k_2\}$, and same applies to all signals like $x_{1:C}$.

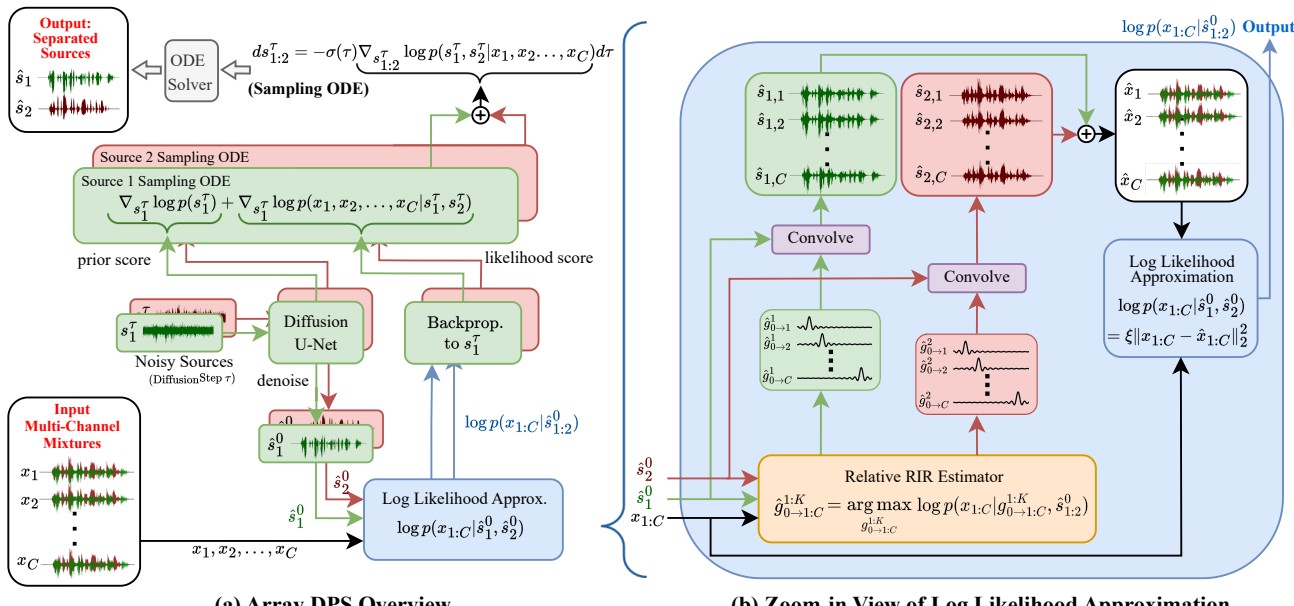

**(a) Array DPS Overview**  **(b) Zoom-in View of Log Likelihood Approximation**

*Figure 2.* An overview of ArrayDPS for $K = 2$ sources. Left figure (a) shows the pipeline for separation with diffusion posterior sampling, which uses the likelihood approximation pipeline shown in the right figure (b).

obtain $s_{k,c_2}(t)$ if $h_{k,c_1}(t)$ is invertible:

$$g_{c_1 \to c_2}^k(t) = h_{k,c_2}(t) * h_{k,c_1}^{-1}(t) \tag{3}$$

$$s_{k,c_2}(t) = g_{c_1 \to c_2}^k(t) * s_{k,c_1}(t) \tag{4}$$

Eq. 3 is called the relative RIR between channel $c_1$ and $c_2$ for speaker $k$, and Eq. 4 is the convolution filtering process. The relative RIR model is useful because the reference channel can serve as an anchor, and all the other channel measurements can be modeled relative to this anchor.

*Can we take one more step and avoid designating a microphone as a reference channel?* One idea is to imagine a virtual source $s_k$ for each speaker and apply a relative RIR filter to map it to a real channel. Fig.1(c) shows an example where relative RIR $g_{0 \to 1}^1(t)$ maps the virtual source $s_1(t)$ to the source image $s_{1,1}(t)$ at channel 1. The same occurs for all ⟨virtual source, real channel⟩ pairs:

$$s_{k,c}(t) = g_{0 \to c}^k(t) * s_k(t), \quad c \in \{1, 2, ..., C\} \tag{5}$$

*What is the advantage of modeling such a virtual source?* The key reason is that virtual sources (along with their relative RIRs) offer the flexibility to model many system configurations. For instance, a virtual source could be the anechoic speaker's voice signal, or it could be the measurement at a real channel $c_1$, or it could even be the signal at an imaginary microphone $c_0$ placed near the speakers. Whatever the virtual signal represents, the corresponding relative RIR filters can adapt to match the measurements $x_{1:C}$ at the real microphones. This flexibility allows (1) treating the reference channel $c_1$ the same as all other channels, and (2) more freedom in the optimization, allowing better performance in some algorithms (Wang & Watanabe, 2023).

## 3. ArrayDPS

### 3.1. Brief Background and Overview

**Background:** Our problem is formulated as $x_{1:C} = A(s_{1:K}) + n_{1:C}$, where $A$ can be understood as the filter and sum process in Fig. 1(c). Separating the sources can be viewed as sampling from the posterior $p(s_{1:K}|x_{1:C})$. For this, we use Diffusion Posterior Sampling (DPS) (Chung et al., 2023b; Song et al., 2023; 2021) which samples from the posterior by solving a score-based probabilistic flow ordinary differential equation (ODE):

$$ds_{1:K}^\tau = -\sigma(\tau)\nabla_{s_{1:K}^\tau} \log p(s_{1:K}^\tau|x_{1:C})d\tau \tag{6}$$

where $s_k^\tau$ is the noisy version of $s_k$ at diffusion time $\tau$. The ODE is adapted from EDM (Karras et al., 2022), assuming a linear noise schedule ($\sigma(\tau) = \tau$). Score-based diffusion model (Song et al., 2021; Karras et al., 2022) suggests that sampling from $p(s_{1:K}|x_{1:C})$ is basically first initializing the sources $s_{1:K}^{\tau_{\max}} \sim \mathcal{N}(0, \sigma^2(\tau_{\max})I)$, and then getting the separated virtual sources $s_{1:K}$ by solving the ODE in Eq. 6 until $\tau = \tau_{\min} \simeq 0$. We elaborate on the details in Appendix A.

To acquire the posterior score $\nabla_{s_{1:K}^\tau} \log p(s_{1:K}^\tau|x_{1:C})$ in the ODE above, DPS decomposes the posterior score into a prior score and a likelihood score, where the former is estimated from a pre-trained diffusion model, and the latter is approximated based on a tractable likelihood model (also explained in Appendix A). However, in our "blind" source separation problem, the likelihood is intractable, and solving this is at the crux of this paper.

**Overview:** Fig. 2(a) shows the ArrayDPS model. The multi-channel audio mixtures $x_{1:C}$ are inputs (shown at the bottom left of the figure) and the outputs are separated vir-

tual sources from the ODE (shown at the top of the figure). The whole ArrayDPS pipeline essentially develops the components needed for the ODE to reverse the diffusion process. At each diffusion time $\tau$, ArrayDPS can be summarized in five key steps (following Fig. 2(a)):

**(Step 1)**: The final posterior score $\nabla_{s_{1:K}^\tau} \log p(s_{1:K}^\tau | x_{1:C})$ in the ODE is constructed by adding the prior score $\nabla_{s_k^\tau} \log p(s_k^\tau)$ and likelihood score $\log \nabla_{s_k^\tau} p(x_{1:C} | s_{1:K}^\tau)$ for *each* source.

**(Step 2)**: The prior score is estimated using Tweedie's Formula: $\nabla_{s_k^\tau} p(s_k^\tau) = (\hat{s}_k^0 - s_k^\tau)/\sigma^2(\tau)$. The $\hat{s}_k^0$ in the formula, which is the estimate of clean source, is the output of a diffusion denoising U-Net with noisy source $s_k^\tau$ as input.

**(Step 3)**: The likelihood $p(x_{1:C} | \hat{s}_{1:K})$ is intractable because the distortion function $A(\cdot)$ (filtering+mixing) includes the unknown relative RIRs. However, given the relative RIRs, the log likelihood $\log p(x_{1:C} | \hat{s}_{1:K}^0, g_{0\to1:C}^{1:K})$ becomes tractable. This motivates our *log likelihood approximation* model expanded in Fig. 2(b).

Given the measured mixture $x_{1:C}$ and the denoised estimates $\hat{s}_{1:K}^0$ from time $\tau$, the yellow module in Fig. 2(b) estimates the relative RIRs $\hat{g}_{0\to1:C}^{1:K}$ in a maximum likelihood manner (detailed next in Sec. 3.2). Recall these relative RIRs are transfer functions from the virtual sources to the real microphone channels, hence appropriately convolving $\hat{g}_{0\to1:C}^{1:K}$ with the virtual source estimates $\hat{s}_{1:K}$ gives us the *actual* source image estimates $\hat{s}_{1:K,1:C}$ at all the microphone channels. Adding up these actual source image estimates yields the estimated mixtures $\hat{x}_{1:C}$. Finally, based on the Gaussian noise model $n_c(t)$ in Eq. 2, the log likelihood can be directly calculated as $\|x_{1:C} - \hat{x}_{1:C}\|_2^2$, weighted by a signal energy-based scaler $\xi$.

**(Step 4)**: Back in Fig. 2(a), the approximated log likelihood is back-propagated to $s_k^\tau$ to compute the likelihood score.

**(Step 5)**: This likelihood score and the prior score (from step 2) are combined to obtain the posterior score at $\tau$.

Note that Fig. 2(a) is separating the *virtual* sources $s_{1:K}$. Since we want to estimate the source images at the reference channel, namely $s_{1:K,1}$, we convolve with the estimated relative RIRs $\hat{g}_{0\to1}^k$ to obtain the separated source images $\hat{s}_{k,1}$.

### 3.2. Relative RIR Estimation

Since log likelihood $\log p(x_{1:C} | \hat{s}_{1:K}^0, g_{0\to1:C}^{1:K})$ is tractable, we propose to estimate the $g_{0\to1:C}^{1:K}$ in a maximum likelihood manner using mixtures $x_{1:C}$ and denoised sources $\hat{s}_{1:K}^0$:

$$\hat{g}_{0\to1:C}^{1:K} = \underset{g_{0\to1:C}^{1:K}}{\arg\max} \; \log \; p(x_{1:C} | g_{0\to1:C}^{1:K}, \hat{s}_{1:K}^0) \quad (7)$$

However, previous work has shown that it is better to estimate the filters in spectral domain (Saijo et al., 2024; Gannot et al., 2017) instead of in the time domain shown above. Hence, we transform to the Short-Time-Fourier-Transform

(STFT) domain. In STFT domain, Eq. 5 becomes:

$$S_{k,c}(l,f) = G_{0\to c}^k(f) \cdot S_k(l,f) \quad (8)$$

where $S_{k,c}(l,f)$ and $S_k(l,f)$ are the STFT of $s_{k,c}(t)$ and $s_k(t)$, respectively. $G_{0\to c}^k(f)$ is the discrete Fourier transform of $g_{0\to c}^k(t)$, where the FFT size is the same as the STFT.

**Relaxing narrowband assumption:** The above equation is unrealistic for speech signals because it makes a narrowband approximation; it assumes that the length of the relative RIR filter is shorter than the FFT size. In real environments, the relative RIR filter can be in hundreds of milliseconds, while the FFT size is usually only tens of milliseconds. To relax this assumption, we model the relative RIR over multiple time-frames as $G_{0\to c}^k(l,f)$. This multi-frame filter in the STFT domain is then *convolved* with the virtual source as follows:

$$S_{k,c}(l,f) = G_{0\to c}^k(l,f) *_l S_k(l,f) \quad (9)$$

where $l,f$ are frame and frequency index, and $F, P$ are the number of future and past frames of the relative RIR $G_{0\to c}^k(l,f)$, all in STFT domain. Observe that $*_l$ in Eq.9 denotes *convolution* in the frame dimension:

$$G_{0\to c}^k(l,f) *_l S_k(l,f) = \sum_{j=-F}^{P} G_{0\to c}^k(j,f) S_k(l-j,f) \quad (10)$$

Thus, the complete signal model — that connects the virtual sources to the actual mixtures at the microphones — can be modeled as:

$$X_c(l,f) = \sum_{k=1}^{K} G_{0\to c}^k(l,f) *_l S_k(l,f) + N_c(l,f) \quad (11)$$

**ML Estimation:** Under this setting, we intend to estimate relative RIRs $G_{0\to c}^k(l,f)$ in a maximum likelihood manner. We found that the Forward Convolutive Prediction (FCP) method (Wang et al., 2021a;b; Wang & Watanabe, 2023) is able to achieve this. Basically, FCP estimates relative RIRs $G_{0\to c}^k(l,f)$ from the multi-channel mixtures $X_c(l,f)$ and the virtual source estimate $\hat{S}_k(l,f)$ by solving a linear problem:

$$\hat{G}_{0\to c}^k(l,f) = \text{FCP}(X_c(l,f), \hat{S}_k(l,f)) \quad (12)$$

$$= \underset{G_{0\to c}^k(l,f)}{\arg\min} \sum_{l,f} \frac{1}{\hat{\lambda}_c^k(l,f)} \left| X_c(l,f) - G_{0\to c}^k(l,f) *_l \hat{S}_k(l,f) \right|^2 \quad (13)$$

$$\hat{\lambda}_c^k(l,f) = \frac{1}{C} \sum_{c=1}^{C} |X_c(l,f)|^2 + \epsilon \cdot \max_{l,f} \frac{1}{C} \sum_{c=1}^{C} |X_c(l,f)|^2 \quad (14)$$

As shown above, FCP is solving a weighted least squares problem, so it has an analytical solution as in (Wang & Watanabe, 2023). The weight $\hat{\lambda}_c^k(l,f)$ aims to prevent overfitting to high-energy STFT bins, and $\epsilon$ is a parameter to adjust the weight. Note that two important parameters for FCP are $F$ and $P$ from Eq. 10, which control the past and future filter lengths of the relative RIR. We prove FCP is equivalent to the maximum likelihood relative RIR estimator in Appendix B.2 Theorem B.1. Thus, with the FCP estimated filters, we can obtain the estimated source images

at all microphone channels:

$$\hat{S}_{k,c}(l,f) = \hat{G}_{0\to c}^k(l,f) *_l \hat{S}_k(l,f) \tag{15}$$

### 3.3. Posterior Score Approximation

The log likelihood approximation gives us relative RIR estimates and the corresponding source image estimates at each microphone channel. All of these were computed based on clean source estimates denoised by the diffusion model at time $\tau$. Using all the available estimates, ArrayDPS now needs to compute the posterior score, and then run the ODE solver in Eq. 6 to ultimately get samples from $p(s_{1:K}|x_{1:C})$. We describe this process next.

Assume we have a pre-trained diffusion denoising model $D_\theta(s_k^\tau, \sigma(\tau))$ trained on single-channel speech sources; the corresponding score model $S_\theta(s_k^\tau, \sigma(\tau))$ approximates $\nabla_{s_k^\tau} \log p(s_k^\tau)$. Based on this, we present here a novel approximation of the posterior score $\nabla_{s_k^\tau} \log p(s_{1:K}^\tau|x_{1:C})$ to enable sampling using the ODE above. All the derivation, proof, and analysis of our approximation is in Appendix B. Below is our final result, where we first denoise the noisy sources $s_{1:K}^\tau$ and then transform to STFT domain:

$$\hat{s}_k^0 = D_\theta(s_k^\tau, \sigma(\tau)), \quad \hat{S}_k^0 = \text{STFT}(\hat{s}_k^0), \quad k \in \{1, ..., K\} \tag{16}$$

Then we use FCP to estimate all relative RIRs:

$$\hat{G}_{0\to c}^k = \text{FCP}(X_c, \hat{S}_k^0), \quad \hat{S}_{k,c} = \hat{G}_{0\to c}^k *_l \hat{S}_k \tag{17}$$

The relative RIRs enable log likelihood estimates (at the output of Fig. 2(b)), and to get their respective likelihood scores, we back-propagate to $s_1^\tau, s_2^\tau$ to calculate the gradient. In the final step, the likelihood scores and the prior scores are summed up to give the posterior score:

$$\nabla_{s_{1:K}^\tau} \log p(s_{1:K}^\tau|x_{1:C}) \simeq \sum_{k=1}^K S_\theta(s_k^\tau, \sigma(\tau)) +$$
$$\sum_{c=1}^C \nabla_{s_{1:K}^\tau} \xi(\tau) \left\| x_c - \text{ISTFT}\left(\sum_{k=1}^K \hat{S}_{k,c}^0\right) \right\|_2^2 \tag{18}$$

ISTFT above denotes Inverse-Short-Time-Fourier-Transform. Note that the functional relationship between $s_{1:K}^\tau$ and $\hat{S}_{1:K,1:C}^0$ is fully differentiable, as operations $D_\theta$, STFT, ISTFT and FCP are all differentiable. Thus, in the second term in Eq. 18, the likelihood score can be calculated by back-propagating to $s_{1:K}^\tau$. Again, detailed proof of correctness, analysis, and relation to Expectation Maximization is included in Appendix B.

### 3.4. IVA as Initialization

At the early stages of solving the separation ODE, the source estimates $\hat{s}_{1:K}^0$ are extremely inaccurate, which means the estimated relative RIRs are inaccurate. To mitigate this, we propose to use an Independent Vector Analysis algorithm (Kim et al., 2006; Hiroe, 2006) as initialization, for both the sources and the relative RIRs.

IVA is a classic unsupervised algorithm for multi-channel blind speech separation, which has limited performance in reverberant conditions (Kim et al., 2006; Hiroe, 2006). For

---

**Algorithm 1** ArrayDPS

**Require:** $\{N, \sigma_{i\in\{0,...,N\}}, \gamma_{i\in\{0,...,N-1\}}, S_{\text{noise}}\}$
**Inputs:** mixtures $x_{1:C}$, number of sources $K$
1: $X_{1:C} \leftarrow \text{STFT}(x_{1:C})$
2: $\hat{S}_{1,1}^{\text{IVA}}, ..., \hat{S}_{K,1}^{\text{IVA}} \leftarrow \text{IVA}(X_{1:C})$ ▷ IVA separation
3: $\hat{s}_{1,1}^{\text{IVA}}, ..., \hat{s}_{K,1}^{\text{IVA}} \leftarrow \text{ISTFT}(\hat{S}_{1,1}^{\text{IVA}}), ..., \text{ISTFT}(\hat{S}_{K,1}^{\text{IVA}})$
4: **for** $k = 1$ **to** $K$, $c = 1$ **to** $C$ **do**
5: $\quad \hat{G}_{1\to c}^{k,\text{IVA}} \leftarrow \text{FCP}(X_c, \hat{S}_{k,1}^{\text{IVA}})$ ▷ IVA Init. relative RIR
6: **end for**
7: **for** $k = 1$ **to** $K$ **do**
8: $\quad$ Initialize $s_k^{\tau 0} \sim \mathcal{N}(\hat{s}_{k,1}^{\text{IVA}}, \sigma_0^2 I)$ ▷ Diff. Init. with IVA
9: **end for**
10: **for** $i = 0$ **to** $N-1$ **do** ▷ $2^{\text{nd}}$order Heun stochastic sampler
11: $\quad \hat{\sigma}_i \leftarrow \sigma_i + \gamma_i \sigma_i$
12: $\quad$ Sample $\epsilon_i \sim \mathcal{N}(0, S_{noise}^2 I)$
13: $\quad$ **for** $k = 1$ **to** $K$ **do**
14: $\quad\quad \hat{s}_k^{\tau_i} \leftarrow s_k^{\tau_i} + \sqrt{\hat{\sigma}_i^2 - \sigma_i^2}\epsilon_i$ ▷ add stochasticity
15: $\quad$ **end for**
16: $\quad score \leftarrow \text{GET\_SCORE}(\hat{s}_{1:K}^{\tau_i}, x_{1:C}, \hat{G}_{1\to c}^{k,\text{IVA}}, i)$ ▷ Algo.2
17: $\quad d_i \leftarrow -\hat{\sigma}_i \cdot score$
18: $\quad s_{1:K}^{\tau_{i+1}} \leftarrow \hat{s}_{1:K}^{\tau_i} + (\sigma_{i+1} - \hat{\sigma}_i) \cdot d_i$ ▷ $1^{\text{st}}$ order Euler step
19: $\quad$ **if** $\sigma_i \neq 0$ **then** ▷ $2^{\text{nd}}$ order correction
20: $\quad\quad score' \leftarrow \text{GET\_SCORE}(s_{1:K}^{\tau_{i+1}}, x_{1:C}, \hat{G}_{1\to c}^{k,\text{IVA}}, i+1)$
21: $\quad\quad d_i' \leftarrow -\sigma_{i+1} \cdot score'$
22: $\quad\quad s_{1:K}^{\tau_{i+1}} \leftarrow \hat{s}_{1:K}^{\tau_i} + \frac{1}{2}(\sigma_{i+1} - \hat{\sigma}_i) \cdot (d_i + d_i')$
23: $\quad$ **end if**
24: **end for**
25: $\hat{S}_{1:K} \leftarrow \text{STFT}(s_{1:K}^{\tau_N})$
26: **for** $k = 1$ **to** $K$ **do** ▷ transform to channel 1 with FCP
27: $\quad \hat{G}_{0\to 1}^{k,\text{final}} \leftarrow \text{FCP}(X_1, \hat{S}_k)$
28: $\quad \hat{s}_{k,1} \leftarrow \text{ISTFT}(\hat{G}_{0\to 1}^{k,\text{final}} * \hat{S}_k)$
29: **end for**
30: **return** $s_{1:K}^{\tau_N}, \hat{s}_{1:K,1}$ ▷ separated virtual and channel 1 sources

---

initialization purposes, we use IVA to separate the sources at the reference channel ($c_1 = 1$) and then use these separated sources to further estimate the relative RIRs $\hat{G}_{1\to c}^{k,\text{IVA}}$:

$$\hat{S}_{1,1}^{\text{IVA}}, ... \hat{S}_{K,1}^{\text{IVA}} = \text{IVA}(X_{1:C}) \tag{19}$$
$$\hat{G}_{1\to c}^{k,\text{IVA}} = \text{FCP}(X_c, \hat{S}_{k,1}^{\text{IVA}}), \quad k \in \{1, ..., K\}, \quad c \in \{1, ..., C\} \tag{20}$$

Then the IVA separated sources $\hat{S}_{1:K,1}^{\text{IVA}}$ are used to initialize the starting point of posterior sampling, and the estimated relative RIRs $\hat{G}_{1\to c}^{k,\text{IVA}}$ are used to guide the posterior sampling in early stages, which will be discussed next in Sec. 3.5.

### 3.5. Algorithm

Our proposed ArrayDPS is specified in Algorithm 1, which calls the posterior score approximation function, GET_SCORE($\cdot$), presented in Algorithm 2. We intuitively explain blocks of the algorithm below while the details can be found in Appendix C.

**ArrayDPS Algorithm:** The ArrayDPS algorithm in Algorithm 1 needs a few sampler parameters to be predefined: sampling steps and schedule $N$, $\sigma_{i\in\{0,...,N-1\}}$, and $\gamma_{i\in\{0,...,N-1\}}$, $S_{\text{noise}}$, which all follow the stochastic

**Algorithm 2** Posterior Score Approximation

---

**Require:** $\{D_\theta, \sigma_{i \in \{0,\dots,N\}}\}, N_{\text{ref}}, N_{\text{fg}}, \xi_1(\tau), \xi_2(\tau), \lambda$
1: **function** GET_SCORE($s_{1:K}^{\tau_i}, x_{1:C}, \hat{G}_{1 \to c}^{k,\text{IVA}}, i$)
2:     **for** $k = 1$ **to** $K$ **do**
3:         $\hat{s}_k^0 \leftarrow D_\theta(s_k^{\tau_i}, \sigma_i)$            ▷ diffusion denoiser
4:         $\hat{S}_k^0 \leftarrow \text{STFT}(\hat{s}_k^0)$
5:     **end for**
6:     $c_i \leftarrow \frac{\hat{s}_{1:K}^0 - s_{1:K}^{\tau_i}}{\sigma_i^2}$             ▷ get prior score
7:     $X_{1:C} \leftarrow \text{STFT}(x_{1:C})$
8:     **for** $k = 1$ **to** $K$, $c = 1$ **to** $C$ **do**
9:         **if** $i \leq N_{\text{fg}}$ **then**
10:             $\hat{G}_{0 \to c}^k \leftarrow \text{FCP}(X_c, \hat{S}_k^0)$     ▷ Est. relative RIR
11:         **else**
12:             $\hat{G}_{0 \to c}^k \leftarrow \hat{G}_{1 \to c}^{k,\text{IVA}}$    ▷ Use IVA init. relative RIR
13:         **end if**
14:         $\hat{s}_{k,c} \leftarrow \text{ISTFT}(\hat{G}_{0 \to c}^k *_l \hat{S}_k^0)$   ▷ filter to all channels
15:     **end for**
16:     $g_i \leftarrow -\xi_1(\tau_i) \nabla_{s_{1:K}^\tau} \sum_{c=1}^C \left\| x_c - \sum_{k=1}^K \hat{s}_{k,c} \right\|_2^2$
17:     **if** $i \leq N_{\text{ref}}$ **then**       ▷ reference channel guidance
18:         $r_i \leftarrow -\xi_2(\tau_i) \nabla_{s_{1:K}^\tau} \left\| x_1 - \sum_{k=1}^K \hat{s}_k^0 \right\|_2^2$
19:         $g_i \leftarrow g_i + \lambda r_i$       ▷ update posterior score
20:     **end if**
21:     **return** $c_i + g_i$         ▷ final posterior score
22: **end function**

---

sampler in EDM (Karras et al., 2022). The algorithm then takes the mixtures $x_{1:C}$ and number of sources $K$ as inputs, to separate source images in the reference channel. From lines 1-6, IVA is used to separate reference-channel source images, which are further used to estimate the relative RIRs. Then in lines 7-9, the IVA separated sources are used as diffusion initialization. Line 10-24 is a second-order Heun stochastic sampler following EDM (Karras et al., 2022), where the score is using our Posterior Score Approximation, GET_SCORE($\cdot$), in Algorithm 2. Note that the sampler is sampling the virtual sources instead of reference channel source images; we discuss the reason later in Appendix G. Then, to output separated source images in the reference channel, lines 25-30 use FCP to estimate the relative RIRs that allow filtering virtual sources to reference channel source images.

**Posterior Score Approximation Algorithm:** The posterior score approximation in Algorithm 2 follows the result in Sec. 3.3 and provides Algorithm 1 with the posterior score for diffusion sampling. Algorithm 2 needs a few pre-defined parameters. $D_\theta$ is pre-trained diffusion denoiser. $N_{\text{ref}}$ is the number of early steps needed for reference channel guidance, where the sum of estimated sources are directly guided by the reference channel mixture (lines 17-19). We found this empirical term helps improve algorithm robustness and convergence. $N_{\text{fg}}$ means in the first $N_{\text{fg}}$ sampling steps, the IVA initialized relative RIRs are used to calculate the likelihood, instead of the FCP-estimated ones. $\xi_1(\tau), \xi_2(\tau), \lambda$ are all weighting scalars which will be discussed in detail

in Appendix C. The algorithm takes the current diffusion noisy sources $s_{1:K}^{\tau_i}$, the mixtures $x_{1:C}$, the IVA initialized relative RIRs $\hat{G}_{1 \to c}^{k,\text{IVA}}$ and the diffusion step $i$ as inputs. In lines 2-6, the noisy sources are denoised, and are then used to estimate the prior score (using Tweedie's Formula). Then in lines 9-15, the relative RIRs (estimated from IVA init. or estimated using current source est.) are used to transform estimated virtual sources to all channels. In line 16 the approximated likelihood score is estimated. Line 17-19 is the reference channel guidance mentioned. Then in line 21, the posterior score is the sum of the prior score and likelihood score.

## 4. Experiments and Evaluation

We first discuss ArrayDPS's training and sampling configurations, as well as the baseline configurations. Then we show results and analysis on both **fixed** microphone array (test samples are all from one single microphone array) and **ad-hoc** microphone arrays (test samples are from different unknown microphone arrays). We have open sourced ArrayDPS in https://github.com/ArrayDPS/ArrayDPS.

### 4.1. Datasets

As mentioned in Sec. 3, ArrayDPS relies on a pre-trained single-channel single-speaker denoising diffusion model $D_\theta$. We train this unconditional speech diffusion model on a clean subset of speech corpus LibriTTS (Zen et al., 2019). Since the virtual source could be reverberant, we train another diffusion prior to reverberant speech, where we convolve LibriTTS clean speech with room impulse responses (RIR). We discuss details, including the architecture and diffusion training configurations, in Appendix D. For the final sampler and the posterior score approximation, we discuss the parameter configurations in Appendix C.3.

For evaluation, we use SMS-WSJ (Drude et al., 2019b) dataset for fixed microphone array evaluation and use Spatialized WSJ0-2Mix dataset (Wang et al., 2018) for ad-hoc microphone array evaluation. SMS-WSJ contains simulated mixtures and sources on a fixed microphone array while Spatialized WSJ0-2Mix contains samples on different microphone arrays. All datasets are designed for 2-speaker source separation. Details of the datasets are explained in Appendix E.

### 4.2. ArrayDPS and Variants (for ablations)

Row 2a–2e, 3a–3b, and 4a–4b in Table 1 show Array-DPS and its variants for ablations. **ArrayDPS-A** in row **2a** is default ArrayDPS with anechoic clean speech diffusion prior. **ArrayDPS-B** in row **2b** is the same but the diffusion prior is trained on reverberant speech. **ArrayDPS-C** in row **2c** shows the ablation when the reference channel guidance is removed from Algorithm 2. **ArrayDPS-D** in row **2d** shows the ablation when IVA initialization is re-

*Table 1.* Evaluation results for 3-channel SMS-WSJ. Methods denoted with † are results from UNSSOR (Wang & Watanabe, 2023), and methods denoted with ∗ mean it is impractical. Note that SMS-WSJ only contains samples with a fixed microphone array. Top results are color-coded as top1 , top2 , and top3 .

| Row | Methods | Unsup. | Array Agnostic | Prior | IVA Init. | Ref. Guide. | SDR (dB) | SI-SDR (dB) | PESQ | eSTOI |
|---|---|---|---|---|---|---|---|---|---|---|
| 0 | Mixture | - | - | - | - | - | 0.1 | 0.0 | 1.87 | 0.603 |
| 1a | Spatial Cluster† | ✓ | ✓ | - | - | - | 9.5 | 8.5 | 2.52 | 0.759 |
| 1b | IVA† | ✓ | ✓ | Laplace | - | - | 12.0 | 10.7 | 2.67 | 0.802 |
| 1c | IVA | ✓ | ✓ | Gaussian | - | - | 13.4 | 12.2 | 2.82 | 0.834 |
| 1d | UNSSOR† | ✓ | ✗ | - | - | - | 15.4 | 14.4 | 3.20 | 0.875 |
| 2a | ArrayDPS-A | ✓ | ✓ | Anechoic | ✓ | ✓ | 15.8±1.2 | 15.0±1.3 | 3.38±0.12 | 0.865±0.020 |
| 2b | ArrayDPS-B | ✓ | ✓ | Reverb. | ✓ | ✓ | 15.1±1.1 | 14.3±1.2 | 3.29±0.11 | 0.850±0.019 |
| 2c | ArrayDPS-C | ✓ | ✓ | Anechoic | ✓ | ✗ | 14.9±1.3 | 14.0±1.5 | 3.16±0.16 | 0.844±0.026 |
| 2d | ArrayDPS-D | ✓ | ✓ | Anechoic | ✗ | ✓ | 8.5±4.6 | 6.8±5.2 | 2.59±0.48 | 0.731±0.111 |
| 2e | ArrayDPS-E | ✓ | ✓ | Anechoic | ✗ | ✗ | 0.9±1.6 | -1.4±1.9 | 1.74±0.15 | 0.518±0.058 |
| 3a | ArrayDPS-A-Max∗ | ✓ | ✓ | Anechoic | ✓ | ✓ | 17.2 | 16.5 | 3.52 | 0.888 |
| 3b | ArrayDPS-D-Max∗ | ✓ | ✓ | Anechoic | ✗ | ✓ | 14.4 | 13.4 | 3.19 | 0.860 |
| 4a | ArrayDPS-A-ML | ✓ | ✓ | Anechoic | ✓ | ✓ | 16.9 | 16.2 | 3.49 | 0.884 |
| 4b | ArrayDPS-D-ML | ✓ | ✓ | Anechoic | ✗ | ✓ | 14.0 | 12.9 | 3.15 | 0.851 |
| 5a | TF-GridNet-SMS† | ✗ | ✗ | - | - | - | 16.8 | 16.3 | 3.91 | 0.924 |
| 5b | TF-GridNet-SMS | ✗ | ✗ | - | - | - | 16.2 | 15.7 | 3.72 | 0.908 |
| 5c | TF-GridNet-Spatial | ✗ | (3-mics) | - | - | - | 14.7 | 14.1 | 3.35 | 0.877 |

moved, while **ArrayDPS-E** in row **2e** shows the ablation when both initialization and reference channel guidance are removed. Note that the sampling parameters for ArrayDPS with and without initialization are different (see Appendix C.3). Because all ArrayDPS in Row 2(a–e) are generative, we sample 5 separated samples for each mixture and report the 5-sample mean and standard deviation (mean±std in Row 2, Table 1), averaged over all the mixtures in the test set.

In Row **3a–3b**, **ArrayDPS-A-Max** and **ArrayDPS-D-Max** shows the max metric score from the 5 samples, averaged for all test samples. Row **4a-4b** represents **ArrayDPS-A-ML** and **ArrayDPS-D-ML**, where ML stands for 'maximum likelihood'. Basically, from the 5 samples, we find the one with the maximum likelihood (highest mixture reconstruction SNR) and report the separation metric averaged over the test set. Row 3a and 4a's methods are the same as the default 2a (default), while 3b and 4b are the same as 2d (no initialization). Note that Row 4 is a practical algorithm because it only uses the mixture to calculate the likelihood.

### 4.3. Baseline Methods

For the baseline models, we include three unsupervised models: **Spatial Clustering** (Rickard, 2007; Tran Vu & Haeb-Umbach, 2010; Sawada et al., 2011; Ito et al., 2016), **Independent Vector Analysis (IVA)** (Kim et al., 2006; Hiroe, 2006), **UNSSOR** (Wang & Watanabe, 2023), as well as supervised TF-GridNet (Wang et al., 2023). **TF-GridNet-SMS†** is the supervised baseline model reported in UNSSOR (Wang & Watanabe, 2023) trained on SMS-WSJ Corpus (Drude et al., 2019b). **TF-GridNet-SMS** and **TF-GridNet-Spatial** are our reproduced TF-GridNet trained on SMS-WSJ and Spatialized WSJ0-2Mix corpus respectively.

Note that TF-GridNet-Spatial in Table 1 row **5c** should generalize to any 3-channel ad-hoc microphone arrays. All baseline models are discussed in detail in Appendix F.

### 4.4. Metrics

For metrics, we use the first channel clean source images as the reference signals. We use the signal-to-distortion ratio (SDR) (Vincent et al., 2006) and Scale-Invariant SDR (SI-SDR) (Roux et al., 2018) to measure sample-level separation performance. We use perceptual evaluation of speech quality (PESQ) (Rix et al., 2001) and extended short-time objective intelligibility (eSTOI) (Taal et al., 2011) to measure perceptual quality and speech intelligibility respectively.

### 4.5. Fixed Microphone Array Evaluation

We show the evaluation result for the 3-channel (using first, third, and fifth mics) SMS-WSJ test set in Table 1. 6-channel SMS-WSJ results are in Appendix I.

**Against Unsupervised:** We first compare against the unsupervised methods. Comparing rows **2a vs. 1(a–d)**, we see ArrayDPS-A's mean score consistently outperforms spatial clustering and IVA-based methods by a substantive margin; also outperforms the unsupervised state-of-the-art method UNSSOR in all metrics except eSTOI. Moreover, ArrayDPS-A's standard deviation is quite robust. Note that UNSSOR is trained only for a fixed array on the SMS-WSJ dataset. If we further compare row **4a vs. 1d**, we see that by selecting the maximum likelihood sample, ArrayDPS-A-ML shows strong improvement over UNSSOR (1.8 dB of SI-SDR, 0.29 PESQ).

**Against Supervised:** Outperforming recent supervised

methods is extremely challenging and all previous unsupervised methods exhibit a gap to supervised methods (Wang & Watanabe, 2023; Saijo et al., 2024). On comparing row **2a vs. 5a–5b**, ArrayDPS-A is consistently worse than the recent supervised methods, as expected. However, comparing row **3a vs. 5a–5b**, ArrayDPS-A-Max shows better SDR and SI-SDR than the supervised method, which means if we sample 5 samples using ArrayDPS, one of the samples can outperform supervised methods in terms of SDR and SI-SDR. However, since ArrayDPS-Max's max operation is not practical, we check ArrayDPS-A-ML, i.e., row **4a vs. 5a–5b**; observe that the maximum likelihood sample from 5 samples achieves slightly better SDR than the supervised method, but a bit worse on other metrics.

**Ablation Studies:** We compare the ablations in rows 2a-2e. Comparing **2b vs. 2a**, we find that it is better to use the anechoic clean speech diffusion prior instead of the reverberant one. The possible reason is related to FCP, which we explain in Appendix G. Comparing **2c vs. 2a**, we find that with initialization, the reference channel guidance can improve all metrics and reduce the standard deviation (instability). Comparing **2d vs. 2a**, the performance drops severely without the IVA initialization, while the std is also high, meaning sometimes the method can separate but not always. When it's able to separate, from row **3b** and **4b**, the best result from 5 samples is reasonable even without IVA initialization. Lastly, we compare row **2d vs. 2e** and find that without IVA initialization, the reference channel guidance is extremely important to obtain reasonable performance.

Lastly, we check row **5c**, which is supervised TF-GridNet trained on 3-channel Spatialized WSJ0-2Mix dataset (ad-hoc array dataset) and we test how well it works on the SMS-WSJ dataset. Observing row **5b vs. 5c**, we see that for supervised models, generalizing to ad-hoc arrays costs a huge performance drop. Comparing row **5b vs. 2a**, ArrayDPS-A outperforms supervised TF-GridNet-Spatial in SDR and SI-SDR by about 1dB, while ArrayDPS-A-ML (row **4a**) outperforms TF-GridNet-Spatial for all metrics.

### 4.6. Ad-hoc Array Evaluation

We evaluate the ad-hoc array setting on the Spatialized WSJ0-2Mix (Wang et al., 2018) dataset. In Table 2, we show results for the 4-channel case (using the first 4 mics for each sample, different samples' array geometries are different); the 2-channel and 3-channel results are in Appendix H.

Following Table 2, ArrayDPS-A-ML (row **5a**) outperforms all *unsupervised* methods by a substantive margin, while ArrayDPS-A (row **2a**)'s mean metrics performs better than UNSSOR (row **1d**) in SDR, SI-SDR, and eSTOI. Comparing row **2a vs. 6a**, we see ArrayDPS-A's mean metric scores are even better than *supervised* TF-GridNet in SDR

*Table 2.* Ad-hoc Array Evaluation results for 4-channel Spatialized WSJ0-2Mix. Note that the microphone positions are random for this dataset. Top results are emphasized in top1 , top2 , and top3 . Methods denoted with ∗ mean it is impractical.

| Row | Methods | SDR | SI-SDR | PESQ | eSTOI |
|---|---|---|---|---|---|
| 0 | Mixture | 0.2 | 0.0 | 1.81 | 0.545 |
| 1a | Spatial Cluster | 9.3 | 8.0 | 2.48 | 0.745 |
| 1b | IVA-Laplace | 7.9 | 5.2 | 2.41 | 0.648 |
| 1c | IVA-Gaussian | 12.5 | 10.1 | 3.01 | 0.808 |
| 1d | UNSSOR | 15.2 | 14.2 | 3.54 | 0.873 |
| 2a | ArrayDPS-A | 16.1 ±0.6 | 15.3 ±0.6 | 3.47 ±0.07 | 0.877 ±0.012 |
| 3a | ArrayDPS-D | 6.7 ±4.5 | 4.7 ±5.0 | 2.45 ±0.51 | 0.677 ±0.120 |
| 4a | ArrayDPS-A-Max* | 16.8 | 16.0 | 3.54 | 0.891 |
| 4b | ArrayDPS-D-Max* | 12.9 | 11.6 | 3.14 | 0.830 |
| 5a | ArrayDPS-A-ML | 16.6 | 15.8 | 3.52 | 0.886 |
| 5b | ArrayDPS-D-ML | 12.3 | 10.8 | 3.06 | 0.810 |
| 6a | TF-GridNet-Spatial | 15.8 | 15.1 | 3.67 | 0.895 |

and SI-SDR, showing ArrayDPS's superiority in ad-hoc array setting.

### 4.7. 3 speakers and Real-World Evaluations

We also show that our ArrayDPS works nicely in 3-speaker and real-world samples. For 3-speaker evaluation, we evaluate ArrayDPS on the Spatialized WSJ0-3Mix dataset, which is the 3-speaker version of the Spatialized WSJ0-2Mix dataset. This means the dataset also contains samples recorded from ad-hoc microphones. The separation results are shown in Appendix J, showing that ArrayDPS easily outperforms supervised methods by a large margin.

For real-world mixture evaluation, we recorded 15 mixtures in a 7m x 4m x 2.7m room. Two volunteers speak simultaneously with weak environmental noise and are recorded by 3 microphones. The separation results are shown in the demo site: https://arraydps.github.io/ArrayDPSDemo/.

### 4.8. Source and Filter Visualizations

Recall that ArrayDPS samples virtual source signals and then uses FCP to estimate the relative RIRs that filter the virtual sources to the reference channel sources. Thus, we conduct a visualization analysis of the virtual sources, and the estimated relative RIRs are in the spectrogram domain. The visualization and a detailed explanation are in Appendix L. The virtual source listening demos are also on our demo site. Our observation is that the virtual source separated is closer to the anechoic clean source speech than the reverberant clean source speech, meaning that ArrayDPS has some dereverberation effects.

## 5. Related Work

Blind speech separation has been advanced greatly by deep neural networks using supervised training (Wang & Chen,

2018; Hershey et al., 2016; Yu et al., 2017; Kalkhorani & Wang, 2024; Wang et al., 2023; Quan & Li, 2023; Taherian et al., 2023; Xu et al., 2024; Li et al., 2024). However, supervised methods have generalization issues (Pandey & Wang, 2020; Zhang et al., 2021), especially in multi-channel scenarios, where training on multiple microphones with array agnostic tricks (Luo et al., 2020; Yoshioka et al., 2022; Taherian et al., 2022) cannot guarantee generalization on unseen microphone arrays (Taherian et al., 2022). Also, supervised methods are usually discriminative, often producing blurred results with nonlinear distortions. To solve this problem, generative separation methods (Subakan & Smaragdis, 2018; Scheibler et al., 2023; Karchkhadze et al., 2024; Lutati et al.) are proposed to sample clean sources conditioned on the given mixture. However, these generative methods still need clean-mixture paired data for supervised training.

For unsupervised separation, independent component analysis (ICA) (Amari et al., 1995; Hyvärinen et al., 2009; Sawada et al., 2019) separates sources by enforcing independence between sources. Independent vector analysis (IVA) (Kim et al., 2006; Hiroe, 2006; Nakashima et al., 2021) adds a Gaussian assumption on each source's STFT bins. Spatial Clustering (Tran Vu & Haeb-Umbach, 2010; Sawada et al., 2011; Ito et al., 2016; Rickard, 2007) based methods separate the sources by clustering the STFT bins using spatial features.

Deep learning based unsupervised source separation has also been studied widely. Mixture invariant training (MixIT) (Wisdom et al., 2020; Tzinis et al., 2020; 2022a;b; Saijo & Ogawa, 2023; Sivaraman et al., 2022) synthesizes new mixtures by mixing real-world mixtures, and train the separation model output sources can be mixed somehow to the original mixtures. MixIT has been modified to the multi-channel scenarios (Wang & Watanabe, 2023; Han et al., 2024), but shows limited performance. Another line of work (Tzinis et al., 2019; Drude et al., 2019a; Seetharaman et al., 2019; Togami et al., 2020; Xu & Choudhury, 2022) uses classic methods like spatial clustering to generate pseudo-labels as training targets. However, the performance is bounded by the classic method. More recently, one group of work tried to exploit the multi-channel signal model for unsupervised training (Aralikatti et al., 2023; Saijo et al., 2024; Wang & Watanabe, 2023; Drude et al., 2019c; Bando et al., 2021; 2022). These methods train a neural network to separate sources, which after filtering and mixing, would be close to the multi-channel mixture. Although these methods show promising results, they do not exploit the speech prior information during training.

One way to incorporate prior is to use a diffusion model (Ho et al., 2020; Karras et al., 2022; Song et al., 2021; Sohl-Dickstein et al., 2015). Score-based diffusion model learns

to generate from a distribution by following a probabilistic SDE/ODE from a noise initialization (Song et al., 2021). Diffusion posterior sampling (DPS) (Chung et al., 2023b; Song et al., 2023; 2021) tries to solve the inverse problem with a pre-trained prior diffusion model and a known likelihood model. Later, a few works are further proposed to solve the blind inverse problem with DPS (Chung et al., 2023a; Moliner et al., 2024b; Bai et al., 2024; Švento et al., 2025b; Thuillier et al., 2025; Švento et al., 2025a; Moliner et al., 2025; Laroche et al., 2024).

More recently, DPS has been applied for source separation (Yu et al., 2023; Mariani et al., 2024; Jayaram & Thickstun, 2020). However, these methods only consider single-channel source separation where the likelihood is tractable, which is not the case for multi-channel problems.

## 6. Conclusion

This paper proposes ArrayDPS, an unsupervised, array-agnostic, and generative method to separate speech from multi-channel mixture recordings. ArrayDPS proposes to use a speech diffusion prior and a novel likelihood approximation to enable posterior sampling. The result shows our method outperforms other unsupervised methods and performs on par with supervised methods in SDR. We leave future work to explore ArrayDPS for more general array inverse problems.

## Acknowledgment

We thank Foxconn and NSF (grant 2008338, 1909568, 2148583, and MRI-2018966) for funding this research. We are also grateful to the reviewers for their insightful feedback.

## Impact Statement

This paper explores advancements in unsupervised, array-agnostic, and generative blind speech separation, which lies in the intersection of several established fields, including array signal processing and machine learning. The algorithm we developed has many positive societal consequences. One notable application of our algorithm is assistive/augmented hearing, which improves communication and accessibility for individuals in noisy environments, especially those with hearing impairments. Also, our method has the potential to improve automatic speech transcription technology on ad-hoc microphone arrays. Although our method is generative, it does not synthesize harmful speeches that are not initially presented in the speech mixture. Nonetheless, the capability to isolate individual voices may pose privacy risks, which need careful regulations. We believe no other concerns require specific emphasis at this point.

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

We organize the appendix as follows:

- Appendix A provides a background on score-based diffusion and diffusion posterior sampling (DPS).

- Appendix B provides proposed posterior score derivation, analysis, and relation to the classic EM algorithm.

- Appendix C presents Algorithm 1 and Algorithm 2 in detail, and provides algorithm configurations.

- Appendix D explains diffusion prior training configuration and model architecture.

- Appendix F discusses all the baseline models in detail.

- Appendix E illustrates the SMS-WSJ and Spatialized WSJ0-2Mix datasets for evaluation.

- Appendix G shows ablation experiments for FCP method and explains why we use the virtual source signal model.

- Appendix H reports the result of 2-channel and 3-channel Spatialized WSJ0-2Mix dataset.

- Appendix I reports the result of 6-channel SMS-WSJ dataset.

- Appendix J reports the 3-speaker separation results on the 4-channel Spatialized WSJ0-3Mix dataset.

- Appendix K shows ArrayDPS's sensitivity and robustness to hyperparameters.

- Appendix L shows the visualization of ArrayDPS separated virtual sources, FCP estimated RIRs, and ArrayDPS separated reference channel sources.

## A. Score-based Diffusion and Diffusion Posterior Sampling

**Score Diffusion Model:** Diffusion-based generative models aim to sample from a data distribution $p_{\text{data}}(s)$, by starting from a noise sample and then gradually denoising it (Song et al., 2021; Karras et al., 2022; Ho et al., 2020). Score-based diffusion model (Song et al., 2021; Karras et al., 2022) can also formulate the diffusion denoising process with a probabilistic flow ODE. Following EDM (Karras et al., 2022), with the specific diffusion noise schedule $\sigma(\tau) = \tau$, where $\tau$ is the diffusion time, then the probabilistic flow ODE is defined by:

$$ds^\tau = -\sigma(\tau)\nabla_{s^\tau} \log p(s^\tau)d\tau \tag{21}$$

where $p(s^\tau) = \mathcal{N}(s, \sigma^2(\tau)I)$. This probabilistic flow ODE allows transformation from $s^{\tau_{\max}}$ (noise) to $s^{\tau_{\min}}$ (data), using an ODE solver like Euler's method. During training, a denoiser $D_\theta(s^\tau, \sigma(\tau))$ is learned to denoise $s^\tau$ with the following denoising loss:

$$\mathbb{E}_\tau \mathbb{E}_{s \sim p_{\text{data}}} \mathbb{E}_{n \sim \mathcal{N}(0, \sigma^2(\tau)I)} \|D_\theta(s + n, \ \sigma(\tau)) - s\|^2 \tag{22}$$

After training, the score function $\nabla_{s^\tau} \log p(s^\tau)$ is approximated by $S_\theta(s^\tau, \sigma(\tau)) =: \frac{D_\theta(s^\tau, \sigma(\tau)) - s^\tau}{\sigma^2(\tau)}$, using the Tweedie's Formula. During sampling, $s^{\tau_{\max}}$ is first sampled from $\mathcal{N}(0, \sigma^2(\tau_{\max})I)$, and then an ODE solver is used to integrate through Eq.21 from $s^{\tau_{\max}}$ to a data sample $s^{\tau_{\min}} \sim p_{\text{data}}$.

**Diffusion Posterior Sampling:** This diffusion model further allows a universal framework to solve the general inverse problem, which is known for diffusion posterior sampling (Chung et al., 2023b; Song et al., 2023). Consider a clean signal $s$ (e.g., image or speech) is distorted by a distortion function $A(\cdot)$, which results in a distorted signal $x = A(s) + \epsilon_n$. To recover the original clean signal $s$, diffusion posterior sampling proposes to sample from $p(s|x)$ by using the probabilistic flow ODE similar to Eq. 21 except using the posterior score:

$$ds^\tau = -\sigma(\tau)\nabla_{s^\tau} \log p(s^\tau|x)d\tau \tag{23}$$

To approximate the posterior score $\nabla_{s^\tau} \log p(s^\tau|x)$, it's decomposed into a prior score and a likelihood score with Bayes' theorem:

$$\nabla_{s^\tau} \log p(s^\tau|x) = \nabla_{s^\tau} \log p(s^\tau) + \nabla_{s^\tau} \log p(x|s^\tau) \tag{24}$$

where $\nabla_{s^\tau} \log p(s^\tau)$ is estimated by $S_\theta(s^\tau, \sigma(\tau))$, and in DPS (Chung et al., 2023b), the second term $\nabla_{s^\tau} \log p(x|s^\tau)$ is empirically approximated by:

$$\nabla_{s^\tau} \log p(x|\hat{s}^0(s^\tau)), \text{ where } \hat{s}^0(s^\tau) = D_\theta(s^\tau, \ \sigma(\tau)) \tag{25}$$

$$= \nabla_{s^\tau} \xi(\tau)\|x - A(D_\theta(s^\tau, \ \sigma(\tau)))\|_2^2 \tag{26}$$

The $\xi(\tau)$ is a parameter to control the amount of likelihood guidance for each step, which theoretically relates to the measurement noise variance. Note that this solution assumes that the distortion function $A(\cdot)$ is known in advance.

# B. Posterior Score Approximation

This section first gives a derivation of the proposed posterior score approximation as in Sec. 3.3 Eq. 18. Then to understand why this makes sense, we give a thorough analysis of our approximation in Sec. B.2, where we also prove FCP as a maximum likelihood estimator. Finally, in Sec. B.3, we connect our likelihood score approximation step with the classic expectation maximization algorithm.

## B.1. Derivation

This section gives a full detailed derivation of our approximation of the posterior score as proposed in Sec. 3.3. Basically, our goal is to approximate $\nabla_{s_{1:K,1}^\tau} \log p(s_{1:K}^\tau|x_{1:C})$ that's needed for sampling. First, similar to Eq. 24, we decompose $\nabla_{s_{1:K}^\tau} \log p(s_{1:K}^\tau|x_{1:C})$ with the Bayes' theorem:

$$\nabla_{s_{1:K}^\tau} \log p(s_{1:K}^\tau|x_{1:C}) = \nabla_{s_{1:K}^\tau} \log p(s_{1:K}^\tau) + \nabla_{s_{1:K}^\tau} \log p(x_{1:C}|s_{1:K}^\tau) \tag{27}$$

$$= \sum_{k=1}^{K} \nabla_{s_k^\tau} \log p(s_k^\tau) + \nabla_{s_{1:K}^\tau} \log p(x_{1:C}|s_{1:K}^\tau) \tag{28}$$

Now notice that in the first part of Eq. 28, all the summands $\nabla_{s_k^\tau} \log p(s_k^\tau)$ (prior score) can be approximated by the score model $S_\theta(s_k^\tau, \sigma(\tau))$ trained on single-speaker speech. Thus, we try to approximate the second part of Eq. 28 (likelihood score):

$$\nabla_{s_{1:K}^\tau} \log p(x_{1:C}|s_{1:K}^\tau) = \sum_{c=1}^{C} \nabla_{s_{1:K}^\tau} \log p(x_c|s_{1:K}^\tau) \tag{29}$$

$$\simeq \sum_{c=1}^{C} \nabla_{s_{1:K}^\tau} \log p(x_c|\hat{s}_1^0(s_1^\tau), ..., \hat{s}_K^0(s_K^\tau)), \quad \text{where } \hat{s}_k^0(s_k^\tau) = D_\theta(s_k^\tau, \sigma(\tau)) \tag{30}$$

Eq. 29 is correct because we assume that for the mixtures, different channels' mixtures are conditionally independent given all the sources, which is our array-agnostic assumption. Eq. 29 to Eq. 30 is based on the likelihood approximation in DPS, as shown before in Eq. 25. However, in Eq. 30, $\log p(x_c|\hat{s}_1^0(s_1^\tau), ..., \hat{s}_K^0(s_K^\tau))$ is still not tractable because the estimated clean sources are single-channel virtual sources, while the mixtures are multi-channel. To build the connection, we transfer to the STFT domain and then estimate the relative RIRs to transform from virtual sources to real channel source images.

First, follow the notation in Sec. 3.2, we denote $X_c$ as the STFT of $x_c$ and denote $\hat{S}_k^0$ as the STFT of $\hat{s}_k^0(s_k^\tau)$. Then we use the FCP algorithm defined in Sec. 3.2 Eq. 12 and Eq. 15 to estimate the relative RIRs which allows the transformation from virtual source estimates to all-channel source images:

$$\hat{G}_{0 \to c}^k = \text{FCP}(X_c, \hat{S}_k^0), \hat{S}_{k,c} = \hat{G}_{0 \to c}^k *_l \hat{S}_k^0, \quad k \in \{1, 2, ..., K\}, \quad c \in \{1, 2, ..., C\} \tag{31}$$

Now given the estimated source images $\hat{S}_{k,c}$, we are able to approximate the likelihood score in Eq. 30 in a tractable way:

$$\nabla_{s_{1:K}^\tau} \log p(x_c|\hat{s}_1^0(s_1^\tau), ..., \hat{s}_K^0(s_K^\tau)) \tag{32}$$

$$= \nabla_{s_{1:K}^\tau} \log p(X_c|\hat{S}_1^0, ..., \hat{S}_K^0) \tag{33}$$

$$\simeq \nabla_{s_{1:K}^\tau} \log p(X_c|\hat{S}_1^0, ..., \hat{S}_K^0, \hat{G}_{0 \to c}^1, ..., \hat{G}_{0 \to c}^K) \tag{34}$$

$$= \nabla_{s_{1:K}^\tau} \xi(\tau) \left\| X_c - \sum_{k=1}^{K} \left( \hat{G}_{0 \to c}^k(l, f) *_l \hat{S}_k^0(l, f) \right) \right\|_2^2 \tag{35}$$

$$= \nabla_{s_{1:K}^\tau} \xi(\tau) \left\| X_c - \sum_{k=1}^{K} \hat{S}_{k,c} \right\|_2^2 \tag{36}$$

$$\simeq \nabla_{s_{1:K}^\tau} \xi(\tau) \left\| x_c - \text{ISTFT} \left( \sum_{k=1}^{K} \hat{S}_{k,c} \right) \right\|_2^2 \tag{37}$$

Eq. 32 to Eq. 33 is just a STFT transform. Eq. 33 to Eq. 34 is an empirical approximation. A simple intuition of this is that Eq. 34 is tractable while Eq. 33 is not. The reason is that the relative RIRs $G_{0 \to c}^k$ are needed for the likelihood to be tractable, according to the signal model in Eq. 11. Thus an empirical choice is to estimate the $G_{0 \to c}^k$ using FCP, as in Eq. 31. Later in next section, we will show that these FCP estimated relative RIRs $\hat{G}_{0 \to 1:C}^{1:K}$ are exactly maximum likelihood relative RIRs:

$$\hat{G}_{0 \to c}^1, .., \hat{G}_{0 \to c}^K = \underset{G_{0 \to c}^1, ..., G_{0 \to c}^K}{\arg\max} \; p(X_c | \hat{S}_1^0, .., \hat{S}_K^0, G_{0 \to c}^1, .., G_{0 \to c}^K) \tag{38}$$

We analyze the validity from Eq. 33 to Eq. 34 (likelihood score approximation) in the next subsection where we also show the connection to the classic Expectation Maximization (EM) algorithm. Eq. 34 to Eq. 35 is based on the signal model in Eq. 11 with the Gaussian measurement noise. Eq. 35 to Eq. 36 is a simplification using Eq. 31, and Eq. 36 to Eq. 37 is based on the power preservation of STFT or the STFT version of Parserval's theorem.

Now we can finalize our derivation of the approximated posterior score using Eq. 28, Eq. 30, and Eq. 37, which derives our result in Eq. 18 Sec. 3.3:

$$\nabla_{s_{1:K}^\tau} \log p(s_{1:K}^\tau | x_{1:C}) \simeq \sum_{k=1}^K S_\theta(s_k^\tau, \sigma(\tau)) + \sum_{c=1}^C \nabla_{s_{1:K}^\tau} \xi(\tau) \left\| x_c - \text{ISTFT}\left(\sum_{k=1}^K \hat{S}_{k,c}\right) \right\|_2^2 \tag{39}$$

### B.2. Analysis and Validation

This section complements the derivation above showing the analysis and validation of our posterior score approximation. We first show that the relative RIR filter estimation using FCP is equivalent to maximum likelihood filter estimation, which would validate Eq. 38. We then validate the likelihood score approximation from Eq. 33 to Eq. 34. Lastly, we show the likelihood score approximation's relationship with the classic Expectation Maximization algorithm.

To show the FCP as a maximum likelihood relative RIR estimator, we first recap the FCP formulation in Eq. 12 and Eq. 13,

$$\hat{G}_{0 \to c}^k(l, f) = \text{FCP}(X_c(l, f), \hat{S}_k(l, f)) = \underset{G_{0 \to c}^k(l,f)}{\arg\min} \sum_{l,f} \frac{1}{\hat{\lambda}_c^k(l, f)} \left| X_c(l, f) - G_{0 \to c}^k(l, f) * \hat{S}_k(l, f) \right|^2 \tag{40}$$

**Theorem B.1.** Assume $X_c(l, f) = \sum_{k=1}^K G_{0 \to c}^k(l, f) *_l S_k(l, f) + N_c(l, f)$, where $N_c(l, f) \sim \mathcal{N}(0, \sigma_N^2 I)$. Then when $\hat{\lambda}_c^k(l, f) = \frac{1}{2\sigma_N^2}$, the FCP relative RIR estimator is a maximum likelihood filter estimator:

$$FCP(X_c, \hat{S}_1^0), ..., FCP(X_c, \hat{S}_K^0) = \underset{G_{0 \to c}^1, ..., G_{0 \to c}^K}{\arg\max} \; \log p(X_c | G_{0 \to c}^1, .., G_{0 \to c}^K, \hat{S}_1^0, .., \hat{S}_K^0) \tag{41}$$

*Proof.* First, let $\hat{G}_{0 \to c}^1, .., \hat{G}_{0 \to c}^K$ be the maximum likelihood estimator as shown below:

$$\hat{G}_{0 \to c}^1, .., \hat{G}_{0 \to c}^K = \underset{G_{0 \to c}^1, .., G_{0 \to c}^K}{\arg\max} \; \log p(X_c | G_{0 \to c}^1, .., G_{0 \to c}^K, \hat{S}_1^0, .., \hat{S}_K^0) \tag{42}$$

$$= \underset{G_{0 \to c}^1, .., G_{0 \to c}^K}{\arg\max} \; \log \mathcal{N}(X_c; \sum_{k=1}^K G_{0 \to c}^K *_l \hat{S}_K^0, \sigma_N^2 I) \tag{43}$$

$$= \underset{G_{0 \to c}^1, .., G_{0 \to c}^K}{\arg\min} \; \frac{1}{2\sigma_N^2} \left\| X_c - \sum_{k=1}^K G_{0 \to c}^K *_l \hat{S}_K^0 \right\|_2^2 \tag{44}$$

Eq. 42 to Eq. 44 is based on the signal model in the assumption. Then based on the orthogonal principle in estimation theory, independence of the sources, and linearity of the filtering operation, Eq. 44 is equivalent to:

$$\hat{G}_{0 \to c}^k = \underset{G_{0 \to c}^k}{\arg\min} \frac{1}{2\sigma_N^2} \left\| X_c - G_{0 \to c}^k *_l \hat{S}_k^0 \right\|_2^2 \quad , \forall k \in \{1, 2, ..., K\} \tag{45}$$

which is exactly the FCP objective mentioned in Eq. 40, where $\hat{\lambda}_c^k(l, f) = \frac{1}{2\sigma_N^2}$ as in the assumption. $\square$

With Theorem B.1, it is also obvious that FCP is equivalent to the maximum posterior solution of the relative RIR because we do not assume any prior of the relative RIR filter. Note that empirically in FCP, the weight $\hat{\lambda}_c^k(l, f)$ is set to $\hat{\lambda}_c^k(l, f) = \frac{1}{C} \sum_{c=1}^{C} |X_c(l, f)|^2 + \epsilon \cdot \max_{l,f} \frac{1}{C} \sum_{c=1}^{C} |X_c(l, f)|^2$ for better regularization.

Now with Theorem B.1 relating FCP with filter likelihood, we further analyze the correctness of the likelihood score approximation from Eq. 33 to Eq. 34 in the derivation section B.1. From Eq. 33 to Eq. 34, we have the FCP estimated relative RIRs so that the likelihood is tractable:

$$\nabla_{s_{1:K}^\tau} \log p(X_c|\hat{S}_1^0, ..., \hat{S}_K^0) \simeq \nabla_{s_{1:K}^\tau} \log p(X_c|\hat{S}_1^0, ..., \hat{S}_K^0, \hat{G}_{0\to c}^1, ..., \hat{G}_{0\to c}^K) \tag{46}$$

Now we find the assumption that would make the approximation into equality:

$$p(X_c|\hat{S}_1^0, ..., \hat{S}_K^0) = \int_{G_{0\to c}^1, ..., G_{0\to c}^K} p(G_{0\to c}^1, ..., G_{0\to c}^K) p(X_c|\hat{S}_1^0, ..., \hat{S}_K^0, G_{0\to c}^1, ..., G_{0\to c}^K) dG_{0\to c}^1, ..., dG^{K,\tau} \tag{47}$$

$$= p(X_c|\hat{S}_1^0, ..., \hat{S}_K^0, \hat{G}_{0\to c}^1, ..., \hat{G}_{0\to c}^K) \quad \text{if } p(X_c|\hat{S}_1^0, ..., \hat{S}_K^0, G_{0\to c}^1, ..., G_{0\to c}^K) = \begin{cases} 1 & \text{if } G_{0\to c}^k = \hat{G}_{0\to c}^k \text{ for all } k \\ 0 & \text{else} \end{cases} \tag{48}$$

We can see that the approximation becomes equality under the assumption that $p(X_c|\hat{S}_1^0, ..., \hat{S}_K^0, G_{0\to c}^1, ..., G_{0\to c}^K)$ is a delta function of all the relative RIRs $G_{0\to c}^1, ..., G_{0\to c}^K$, where the function has non-zeros values only when the relative RIRs are the FCP estimated relative RIRs $\hat{G}_{0\to c}^1, ..., \hat{G}_{0\to c}^K$, which are also the maximum likelihood estimators as mentioned before. This assumption makes sense when there is no measurement noise, where a slight error in the relative RIR would cause a likelihood of 0. However, of course, when there is relatively larger measurement noise, the assumption is no longer true, but would still work empirically as shown in our SMS-WSJ separation result in Table 1.

## B.3. Likelihood Approximation and Expectation Maximization

While we analyze the correctness of our posterior score approximation in the previous section, we also find that the most challenging step (likelihood score approximation Eq. 46) has a strong connection with the classic Expectation Maximization algorithm. Note that in EM algorithm, when the likelihood $p(X_c|\hat{S}_1^0, ..., \hat{S}_K^0)$ is intractable because of the unobserved relative RIRs $G_{0\to c}^1, ..., G_{0\to c}^K$, the Expectation (E) step and the Maximization (M) step are iterated:

$$\text{E-Step: calculate } \psi(G_{0\to c}^1, ..., G_{0\to c}^K) \leftarrow p(G_{0\to c}^1, ..., G_{0\to c}^K|X_c, \hat{S}_1^0, ..., \hat{S}_K^0) \tag{49}$$

$$\text{M-Step: update } \hat{S}_1^0, ..., \hat{S}_K^0 \leftarrow \underset{\hat{S}_1^0, ..., \hat{S}_K^0}{\arg\max} \mathbb{E}_{\psi(G_{0\to c}^1, ..., G_{0\to c}^K)} \left[ \log p(X_c|\hat{S}_1^0, ..., \hat{S}_K^0, G_{0\to c}^1, ..., G_{0\to c}^K) \right] \tag{50}$$

The E step updates the distribution $\psi(G_{0\to c}^1, ..., G_{0\to c}^K)$ of the unobserved relative RIRs to be the distribution of the relative RIRs given the current source estimates $\hat{S}_1^0, ..., \hat{S}_K^0$ and the mixture $X_c$. Then using the E step updated distribution $\psi(G_{0\to c}^1, ..., G_{0\to c}^K)$, the M step maximizes the expectation of log likelihood by update the source estimates $\hat{S}_1^0, ..., \hat{S}_K^0$. Then this procedure is iterated until convergence, which is called the Expectation Maximization algorithm.

In our case, we only want to estimate the current log likelihood without updating the sources estimate $\hat{S}_1^0, ..., \hat{S}_K^0$, so our log likelihood approximation only contains one E step and then calculate the expected log likelihood as in the M step (Eq.50). However, The E step is also not tractable in our case, so we just approximate it with:

$$\psi(G_{0\to c}^1, ..., G_{0\to c}^K) \simeq \begin{cases} 1 & \text{if } G_{0\to c}^k = \hat{G}_{0\to c}^k \text{ for all } k \\ 0 & \text{else} \end{cases} \tag{51}$$

$$\text{where } \hat{G}_{0\to c}^1, \hat{G}_{0\to c}^{2,\tau}, ..., \hat{G}_{0\to c}^K = \underset{G_{0\to c}^1, ..., G_{0\to c}^K}{\arg\max} \log p(G_{0\to c}^1, ..., G_{0\to c}^K|X_c, \hat{S}_1^0, ..., \hat{S}_K^0) \tag{52}$$

$$= \underset{G_{0\to c}^1, ..., G_{0\to c}^K}{\arg\max} \log p(X_c|G_{0\to c}^1, ..., G_{0\to c}^K, \hat{S}_1^0, ..., \hat{S}_K^0) \tag{53}$$

$$= \text{FCP}(X_c, \hat{S}_1^0), \text{ FCP}(X_c, \hat{S}_2^0), ..., \text{ FCP}(X_c, \hat{S}_K^0) \tag{54}$$

We set the distribution $\psi(G_{0\to c}^1, ..., G_{0\to c}^K)$ in the E step to be deterministic at $\hat{G}_{0\to c}^1, ..., \hat{G}_{0\to c}^K$, which are both the maximum likelihood filters and maximum posterior filters because no priors are assumed for the filters. Then as mentioned in Sec. B.2

Theorem B.1, $\hat{G}^1_{0 \to c}, ..., \hat{G}^K_{0 \to c}$ can be directly estimated by the FCP method. Then after the E step, we can calculate the expected log likelihood in M step by only calculating $\log p(X_c | \hat{S}^0_1, ..., \hat{S}^0_K, \hat{G}_{0 \to c}, ..., \hat{G}^K_{0 \to c})$, because $\psi$ is defined to be a deterministic distribution.

## C. Algorithm Details

This section provides all the details of the two algorithms in Sec. 3.5, including ArrayDPS as in Algorithm 1 and posterior score approximation in Algorithm 2.

### C.1. ArrayDPS Algorithm

This section discusses the ArrayDPS Algorithm 1 in detail.

**IVA Initialization:** From lines 1-6 in Algorithm 1, IVA first separates $K$ speech source images in the reference channel (channel 1). Then these IVA-separated sources are used as source estimates to calculate the relative RIR using FCP. Note that in lines 4-6, the relative RIR $\hat{G}^{\text{IVA}}_{1 \to c}$ is calculated for all channels. These IVA-initialized relative RIRs will be later used for posterior score approximation as in Algorithm 2, which will be discussed later. Then line 7-9 initializes the starting diffusion noise with the IVA separated sources, in the reference channel (channel 1). Although we are trying to output a virtual source, we still initialize it to be the reference channel's IVA outputs.

**Stochastic Sampler:** Lines 10-24 in Algorithm 1 show the discrete sampler that integrates through the separation ODE in Eq. 6 starting from the diffusion noise initialized in line 8. This sampler is proposed in EDM (Karras et al., 2022), with the parameters $N, \sigma_{i \in \{0,...,N-1\}}, \gamma_{i \in \{0,...,N-1\}}, S_{\text{noise}}\}$ and inputs $\{x_{1:C}, K\}$. $x_{1:C}$ is the multi-channel mixture input and $K$ is the number of sources to separate. $N$ is the number of diffusion denoising steps. $\sigma_{i \in \{0,...,N\}}$ is the noise schedule for each diffusion denoising step (i.e., amount of noise to remove at each step), where $\sigma_N = 0, \sigma_0 = \sigma(\tau_{max}), \sigma_{N-1} = \sigma(\tau_{\min})$. Then all other steps' schedule follows:

$$\sigma_{i<N} = \left( \sigma(\tau_{\max})^{\frac{1}{\rho}} + \frac{i}{N-1} \left( \sigma(\tau_{\min})^{\frac{1}{\rho}} - \sigma(\tau_{\max})^{\frac{1}{\rho}} \right) \right)^{\rho} \tag{55}$$

$\rho$ is a parameter that controls how the noise level per step changes from $\sigma(\tau_{max})$ to $\sigma(\tau_{min})$, e.g., $\rho > 1$ means the noise level per step will decrease more rapidly as $i$ increases. Note that in our ODE settings as in Sec.3, $\sigma(\tau_i) = \tau_i = \sigma_i$. $\gamma_{i \in \{1,...,N-1\}}$ are parameters to control the amount of stochasticity added in the beginning of each step, as in lines 11-14. In EDM (Karras et al., 2022), $\gamma_{i \in \{1,...,N-1\}}$ is set to be:

$$\gamma_i = \begin{cases} \min\left( \frac{S_{\text{churn}}}{N}, \sqrt{2} - 1 \right) & \text{if } \sigma_i \in [S_{\min}, S_{\max}] \\ 0 & \text{otherwise} \end{cases} \tag{56}$$

, where $S_{\text{churn}}$ controls the amount of stochasticity and $S_{\min}, S_{\max}$ set the steps for adding stochasticity. $S_{\text{noise}}$ in line 12 also controls the amount of stochasticity within step $i \in [S_{\min}, S_{\max}]$. In general, the sampler first has a usual $1^{\text{st}}$ order Euler step to update the sources (16-18), and then the newly updated source is used to calculate a $2^{\text{nd}}$ order correction step (lines 20-22). More details of the sampler can be checked in the original EDM paper (Karras et al., 2022).

**Post FCP Filtering:** Note the sampler mentioned above in Algorithm 3.5 is separating virtual sources (line 24). To get the final separated source images in reference channel 1, post-filtering is needed. Lines 26-29 use the FCP algorithm to estimate the relative RIR that can transform the virtual sources into the reference channel. Then after convolutional filtering (line 28), the final reference channel-separated source images are returned as outputs (line 30).

### C.2. Posterior Score Approximation Algorithm

In Algorithm 1, each update step needs to calculate the posterior score, which uses the GET_SCORE function in Algorithm 2. The function GET_SCORE($s^{\tau_i}_{1:K}, x_{1:C}, \hat{G}^{k,\text{IVA}}_{1 \to c}, i$) aims to approximate the posterior score $\nabla_{s^\tau_{1:K}} \log p(s^\tau_{1:K} | x_{1:C})$ at sampling step $i$, using our propose approximation formulation in Sec. 3.3, but with some empirical modifications. The function needs a few pre-defined parameters $\{D_\theta, \sigma_{i \in \{0,...,N\}}, N_{\text{ref}}, N_{\text{fg}}, \xi_1(\tau), \xi_2(\tau)\}$. $D_\theta$ is the diffusion denoiser mentioned in Sec. 3.3. $\sigma_{i \in \{0,...,N\}}$ is the sampler's noise schedule as mentioned in the stochastic sampler paragraph.

$N_{\text{ref}}$ is a sampler step threshold such that when $i \leq N_{\text{ref}}$, there is an empirical reference-channel guidance term that aims to match the sum of estimated virtual sources $\sum^K_{k=1} \hat{s}_k$ to the reference channel (channel 1) mixture $x_1$. This is shown in line

18 in Algorithm. 2. We found that this is important to the algorithm's stability. $N_{\text{fg}}$ is also a sampler step threshold for the filtering guidance of IVA initialized relative RIR. As shown in lines 7-14, when $i \leq N_{\text{fg}}$, instead of using FCP to estimate the relative RIR, the IVA initialized relative RIR $\hat{G}_{1 \to c}^{k,\text{IVA}}$ is directly used. We found that this helps a lot for sampling stability and performance gain. The intuition is that in the first $N_{\text{fg}}$ steps, the source estimate would be extremely blurred and so as the relative RIR estimates, while the IVA initialized relative RIR is a much better choice. Note that $\hat{G}_{1 \to c}^{k,\text{IVA}}$ are actually relative RIRs from reference channel 1 to all channels, but our estimated sources are all virtual sources. We clarify here that this is on purpose, which means in the first $N_{\text{fg}}$ steps, the virtual source estimation would be close to the reference channel, and then the constraint is relaxed in later steps.

$\xi_1(\tau)$ and $\xi_2(\tau)$ are the weights for the mixture likelihood guidance in lines 16 and l8 in Algorithm. 2. Theoretically, $\xi_1(\tau) = \frac{1}{2\sigma_n^2}$, where $\sigma_n^2$ is the white noise variance defined in Eq. 2, and $\xi_2(\tau) = 0$, because the term in line 18 is empirical. However, for sampling stability, we resort to the common practice used in previous audio posterior sampling work (Moliner et al., 2023; 2024a;b), where we set $\xi_1(\tau)$ and $\xi_2(\tau)$ to the following:

$$\xi_1(\tau) = \frac{\xi\sqrt{T}}{\tau \left\| \nabla_{s_{1:K}^\tau} \sum_{c=1}^{C} \left\| x_c - \sum_{k=1}^{K} \hat{s}_{k,c} \right\|_2^2 \right\|_2} \tag{57}$$

$$\xi_2(\tau) = \frac{\xi\sqrt{T}}{\tau \left\| \nabla_{s_{1:K}^\tau} \left\| x_1 - \sum_{k=1}^{K} \hat{s}_k \right\|_2^2 \right\|_2} \tag{58}$$

All the variables in Eq. 57 and Eq. 58 directly refer to the same variables in lines 16 and 18 in Algo. 2. $T$ is the length of the audio samples. $\xi$ is just a single scalar parameter to tune the mixture likelihood guidance. Note that $\xi_1(\tau)$ and $\xi_2(\tau)$ directly depend on the calculated gradient term so the the notations $\xi_1(\tau)$ and $\xi_2(\tau)$ are not accurate. However, we stick to it because the notation is simpler and it is also used in previous papers (Moliner et al., 2023; 2024a).

Overall, in Algorithm 2, the diffusion denoiser first denoises the noisy virtual sources (lines 3), and then the prior score is calculated using Tweedie's formula (line 6). Then depending on the current sample step $i$, the algorithm either chooses the IVA initialized relative RIR (line 12) or directly estimates the relative RIR (line 10) using FCP. The relative RIR is then used to filter the virtual source estimates to all real channels (channel 1 to C) (line 14), which allows the calculation of the gradient likelihood guidance (line 16) and the reference channel guidance (19). Finally, the prior score and the guidance terms are added together to output the approximated posterior score (line 21).

### C.3. Algorithm Configurations

Here we show our ArrayDPS configurations. For the default configuration as in row 2a (ArrayDPS-A) in Table 1, we set $N = 400$ and $S_{\text{noise}} = 1$ as in Algorithm 1, $\sigma_0 = \tau_{\max} = 0.8$, $\sigma_N = \tau_{\min} = 1e-6$, $\rho = 10$ as in Eq. 55, $S_{\min} = 0$, $S_{\max} = 50$, and $S_{\text{churn}} = 30$ as in Eq 56, $\xi = 2$, $N_{\text{ref}} = 200$, $N_{\text{fg}} = 100$, and $\lambda = 1.3$ as in Algorithm 2. The diffusion prior trained on anechoic speech is used. For the FCP algorithm, we use an FFT size of 512 (64 ms), hop size of 64 (8 ms), square root Hanning window, and $\epsilon = 0.001$ as in Eq. 14. For the IVA initialization, we use the open-source *torchiva* toolkit (Scheibler & Saijo, 2022), and we use FFT size of 2048 (256 ms), hop size 256 (32 ms), Gaussian Prior, and 100 iterations.

For ArrayDPS-B in row 2b of Table 1, the diffusion model is trained on reverberant speech, as discussed in Sec. 4.2 and Appendix D. For ArrayDPS-C in row 2c of Table 1, the reference channel guidance in Algorithm 2 is removed, which means $\lambda = 0$. For ArrayDPS-D and ArrayDPS-E in row 2d-2e of Table 1, the IVA initialization is removed, which means $N_{\text{fg}} = 0$ and no IVA is needed. Moreover, in the case of no IVA initialization, we set $\sigma_0 = \tau_{\max} = 2$ and $\xi = 6$ since we found this set of parameters perform better when tuning on validation sets. For ArrayDPS-E, the reference-channel guidance is also removed, so $N_{\text{ref}} = 0$ in this case.

## D. Diffusion Training Details

As mentioned in Sec.4.2, the diffusion prior is trained on the train-clean-100 and train-clean-360 subset of LibriTTS dataset (Zen et al., 2019). This subset contains about 460 hours clean speech with 1000+ speakers. However, this training set is primarily reverberation-free or anechoic. Since the virtual source could be reverberant, we also train a second diffusion

model on reverberant speech. The reverberant speech dataset is the same subset of LibriTTS, but each speech utterance is convolved with a randomly selected room impulse response (RIR) to synthesize reverberant speech. The randomly selected RIRs are from the SLR26 and SLR28 dataset (Ko et al., 2017), which contains 3,076 real and about 115,000 synthetic RIRs.

For the diffusion denoising architecture, we use the waveform domain U-Net as MSDM (Mariani et al., 2024), implemented in audio-diffusion-pytorch/v0.0.432[2]. The U-Net takes audio waveform as inputs, and also outputs audio waveforms. The U-Net consists of encoder, bottleneck, and decoder. The encoder contains 6 layers of 1-D convolutional ResNet blocks, where the last three layers also contain multi-head slef-attention blocks following the ResNet block. The input channel is 1 because we are modeling single channel speech. The number of output channels for each encoder layer is [256, 512, 1024, 1024, 1024, 1024], and the number of downsampling factors for each layer is [4, 4, 4, 2, 2, 2]. For self attention blocks, we use 8 attention heads where each is 128 dimensional. The bottleneck also contains an attention block, with one ResNet block before the attention block and one ResNet block after the attention block. The decoder is then reverse symmetric to the encoder, with output channel size 1 to output a same size waveform.

For diffusion training configurations, we use the EDM (Karras et al., 2022) training framework following the training recipe in CQT-Diffusion[3]. The diffusion denoiser $D_\theta(s^\tau, \sigma(\tau))$ is set to be a linear combination of the noisy input and the U-Net $f_\theta$ output:

$$D_\theta(s^\tau, \sigma(\tau)) = c_{\text{skip}}(\sigma(\tau)) \cdot s^\tau + C_{\text{out}}(\sigma(\tau)) \cdot f_\theta(c_{\text{in}}s^\tau, c_{\text{noise}}) \tag{59}$$

$$\tag{60}$$

,where $c_{\text{skip}}(\sigma)$, $c_{\text{out}}(\sigma)$, $c_{\text{in}}(\sigma)$, $c_{\text{noise}}(\sigma)$, are defined as:

$$c_{\text{skip}}(\sigma) = \frac{\sigma_{\text{data}}^2}{\sigma^2 + \sigma_{\text{data}}^2}, \ c_{\text{out}}(\sigma) = \frac{\sigma \cdot \sigma_{\text{data}}}{\sqrt{\sigma^2 + \sigma_{\text{data}}^2}}, \ c_{\text{in}}(\sigma) = \frac{1}{\sqrt{\sigma^2 + \sigma_{\text{data}}^2}}, \ c_{\text{noise}}(\sigma) = \frac{1}{4}\ln(\sigma) \tag{61}$$

$\sigma_{\text{data}}$ shown above is a parameter to set based on the standard deviation of the training dataset. We set this value to 0.057. Then the diffusion denoising objective is shown as:

$$\mathbb{E}_{\tau \sim p_\tau} \mathbb{E}_{s \sim p_{\text{data}}} \mathbb{E}_{n \sim \mathcal{N}(0, \sigma^2(\tau)I)} \|D_\theta(s + n, \sigma(\tau)) - s\|^2 \tag{62}$$

where $p_\tau$ is empirically set to match the diffusion sampling scheduler's density. During training, we set $\sigma(\tau_{\text{max}}) = 10$, $\sigma(\tau_{\text{min}}) = 1e - 6$, $\rho = 10$, $S_{\text{churn}} = 5$ for the diffusion scheduler. We train on speech samples with 65,536 samples ($\sim$8.2 s) with batsh size 16 and learning rate 0.0001. The learning rate is multiplied by 0.8 every 60,000 training steps. Also exponential moving average (EMA) is used to update the neural network weights during training. The EMA updated weights after training is used for sampling in inference time. We train the model for 840,000 training steps.

## E. Evaluation Datasets

In this paper, two datasets are used for evaluation. The first dataset is SMS-WSJ (Drude et al., 2019b) for **fixed** microphone array evaluation, where all the methods are evaluated on one single microphone array recorded samples. The second dataset is Spatialized WSJ0-2Mix (Wang et al., 2018) for **ad-hoc** microphone array evaluation, where methods are evaluated on multiple unknown microphone arrays.

### E.1. SMS-WSJ Dataset

SMS-WSJ (Drude et al., 2019b) is a commonly used reverberant speech separation corpus (Wang & Watanabe, 2023; Taherian et al., 2024; Saijo et al., 2024; Kalkhorani & Wang, 2024). The dataset is simulated from WSJ0 and WSJ1 datasets. The dataset is for multi-channel 2-speaker separation in reverberant conditions for a fixed simulated microphone array. The simulated microphone array is a circular array with six microphones, uniformly on a circle with 10 cm of radius. For each sample, the shoebox room is randomly sampled, with the length and width uniformly sampled from [7.6, 8.4] m and [5.6, 6.4] m, respectively. The array center position is randomly sampled from [3.6, 4.4] m from the shorter wall and [2.6, 3.4] m from the longer wall. Then the array is randomly rotated along all three geometric axes. The mixture contains two speaker sources, which are sampled form WSJ0 and WSJ1 datasets. For each speaker, the speaker-to-array distance is uniformly

---

[2]https://github.com/archinetai/audio-diffusion-pytorch/tree/v0.0.43
[3]https://github.com/eloimoliner/CQTdiff

sampled from [1.0, 2.0] m, and each speaker position's azimuth related to the array center is also randomly sampled. The room impulse responses are simulated using image-source method (Allen & Berkley, 1979). The sound decay time (T60) is uniformly sampled from [200, 500] ms. Finally, to simulate sensor noise, an SNR value is sampled to be [20, 30] dB, and then the white noise is sampled and scaled to satisfy the SNR required. The dataset consists 33,561 (∼87.4 h), 982 (∼2.5 h), and 1,332 (∼3.4 h) train, validation, and test mixtures, respectively, all in 8kHz sampling rate.

### E.2. Spatialized WSJ0-2Mix Dataset

Spatialized WSJ0-2Mix (Wang et al., 2018) is the spatialized version of the WSJ0-2Mix (Hershey et al., 2016) dataset, which is a commonly used speech separation dataset mixed by randomly selecting sources from the WSJ0 corpus. Multi-channel Deep Clustering (Wang et al., 2018) creates the Spatialized WSJ0-2Mix dataset for multi-channel reverberant speech separation. For each dataset sample in WSJ0-2Mix, a shoebox room, 8-channel microphone array geometry, source positions, and microphone array center are sampled randomly to spatialize that sample. The room length $l$, width $w$, and height are uniformly sampled from [5, 10] m, [5, 10] m, and [3, 4] m, respectively. The RIRs' sound decay time (T60) is sampled from [200, 600] ms. The microphone array center is set to be $(l/2 + n_1, w/2 + n_2, h)$, where $n_1$ and $n_2$ are uniformly sampled from [-0.2, 0.2] m, and $h$ is uniformly sampled from [1, 2] m. Then the microphone array geometry is randomly sampled. First, the array radius is uniformly sampled from [7.5, 12,5] cm. The first two microphones are sampled on the sphere with the sampled radius, and are symmetric according to the array origin. Two more microphones are randomly sampled inside the sphere while making sure the distances between any two microphones has to be at least 5 cm. Then 4 more microphones are randomly sampled inside the sphere without any restrictions. In total, eight microphones are randomly sampled. For the speakers, each speaker's location $(x, y, z)$ is sampled such that $l/2 + n_1 - 1.5 \leq x \leq l/2 + n_1 + 1.5, w/2 + n_2 - 1.5 \leq y \leq w/2 + n_2 + 1.5, 1.5 \leq z \leq 2$. The source locations are sampled such that the source-array distance is at least 0.5 m, and source-source distance is at least 1 m. The RIR simulation is also using the image source method (Allen & Berkley, 1979). The sample rate is set to 8kHz.

In general, the Spatialized WSJ0-2Mix dataset contains 20,000(∼30h), 5,000 (∼10h), and 3,000 (∼5h) utterances in training, validation, and testing, respectively. Different utterances are using different microphone geometries, allowing training models that can generalize to diverse ad-hoc microphone arrays. Also, the direct-to-reverb energy radio of the dataset is 2.5 dB with 3.8 dB of standard deviation.

## F. Baseline Methods

As in Row **1a** of Table 1, Spatial clustering (Rickard, 2007; Tran Vu & Haeb-Umbach, 2010; Sawada et al., 2011; Ito et al., 2016) tries to cluster the multi-channel spatial features in STFT domain. Then each STFT bin is assigned to a source based on the clustering, which relies on the assumption that different speech do not overlap in STFT domain, known as W-Disjoint Orthogonality (W-DO) (Rickard, 2007). We use the same spatial clustering configuration as in the baselines of UNSSOR (Wang & Watanabe, 2023), which uses the open source implementation (Boeddeker, 2019). The FFT size is 1024 (256 ms) and the hop size is 128 (32 ms). The number of sources is set to $K + 1$ and the lowest energy source is discarded as a garbage source.

For the Independent Vector Analysis baseline, similar to the baseline in UNSSOR (Wang & Watanabe, 2023), we use the *torchiva* toolkit with FFT size 2048 (64 ms) and hop size 256 (32 ms). In UNSSOR, the spherical Laplace prior is used as in **Row 1b** of Table 1. We also include Gaussian prior because we find it to perform better, as shown in **Row 1c** of Table 1.

For the UNSSOR baseline trained on Spatialized WSJ0-2Mix datasets, we use the default configuration where both MC and ISMS loss are used, and $I = 19$ and $J = 0$, following the notation in the paper. We use the open source code[4] to train UNSSOR. We found that UNSSOR training is quite sensitive to the ISMS loss weight, which is $\gamma$ in the original paper. Thus we sweep $\gamma \in \{0.03, 0.04, 0.05, 0.06, 0.08, 0.3\}$ and report the best result for the 2/3/4-channel of Spatialized WSJ0-2Mix dataset. For the SMS-WSJ dataset, we just borrow the result in the original UNSSOR paper and report as UNSSOR[†] as in Table 1.

**Row 5a-5c** of Table 1 show the supervised TF-GridNet (Wang et al., 2023) with permutation invariant training (PIT) (Yu et al., 2017), where TF-GridNet-SMS[†] is the exact model trained on SMS-WSJ dataset provided as the supervised baseline of UNSSOR, showing superior performance. However, we also need a supervised baseline model trained on the Spatialized WSJ0-2Mix dataset. Thus we reproduce a supervised TF-GridNet training pipeline using the same architecture with

---

[4]https://github.com/merlresearch/reverberation-as-supervision

TF-GridNet-SMS[†] as in UNSSOR, but trained with with SI-SDR(Roux et al., 2018) PIT loss for simplicity (we are not clear about the training details of the TF-GridNet supervised baseline reported in UNSSOR). We use ADAM optimizer and start with 1e-4 learning rate. We half the learning rate whenever the validation loss do not improve in 3 consecutive epochs. We use batch size 4 and each training segment is 4 seconds long. As in **Row 5b** of Table 1, TF-GridNet-SMS is our implemented supervised TF-GridNet trained on 3-channel SMS-WSJ with 120 epochs (about 1,000,000 steps). In Table 2, TF-GridNet-Spatial is our implementation of the supervised TF-GridNet trained on Spatialized WSJ0-2Mix dataset mentioned in E.2, which tries to generalize to any 3-channel microphone arrays. The model is trained for 200 epochs (about 1,000,000 steps) on Spatialized WSJ0-2Mix dataset. These models are all trained on a single A100 GPU and converges in about 5-6 days.

## G. FCP Ablations

The FCP mentioned in Sec. 3.2 takes a $c_1$ channel mixture $X_{c_1}$ and a $c_2$ channel source estimate $\hat{S}_{k,c_2}$ as inputs, and then output the relative RIR $\hat{G}^k_{c_2 \to c_1}$ that can transform $\hat{S}_{k,c_2}$ to $\hat{S}_{k,c_1}$:

$$\hat{G}^k_{c_2 \to c_1} = \text{FCP}(X_{c_1}, \hat{S}_{k,c_2}), \quad \hat{S}_{k,c_1} = \hat{G}^k_{c_2 \to c_1} *_l \hat{S}_{k,c_2} \tag{63}$$

As discussed in Sec. 2, we decided to use the virtual sources as the estimated sources, which is more flexible comparing with reference channel source images. Here we list a few obvious choices for the virtual source and reason why we use the virtual source based on the oracle FCP performance.

First, the virtual source can be the anechoic source signal $\tilde{S}_k$. Second, the virtual source can be a partially-reverberant source signal $S^{\text{early}}_{k,c_2}$ in channel $c_2$, where only the first 50 milliseconds of the actual room impulse response is applied to filter the anechoic source $\tilde{S}_k$. Note that this early reverberant signal is motivated by the REVERB challenge (Kinoshita et al., 2013), where people found adding the first 50 ms of early reverberation to the anechoic signal sounds better. Third and last, since the virtual source is flexible, it can also be the channel $c_2$'s reverberant image $S_{k,c_2}$.

Now we do a few FCP ablations for different kinds of virtual sources. Without loss of generality, we let $c_1 = 1$. Remember the FCP function always takes two signal inputs: the FCP target signal, and the source estimate signal. In Eq. 63, the FCP target is $X_{c_1}$, and the source estimate signal is the channel $c_2$ estimate $\hat{S}_{k,c_2}$. Here, we do the ablations for different kinds of FCP input combinations, where the source estimate inputs can be the three kinds of virtual sources mentioned above (anechoic, early, reverberant), and the FCP target can be either channel $c_1 = 1$ mixture, or the channel $c_1 = 1$ reverberant source image. Now given the ground-truth virtual sources and the FCP targets, FCP is used to estimate the relative RIRs. Then after applying the estimated relative RIR on the virtual source estimate, we get the channel 1 source image estimate $\hat{S}_{k,1}$ which is used to calculate the speech metrics in Table 3. We calculate all the results in SMS-WSJ test set.

*Table 3.* FCP ablations for different kinds of FCP inputs combinations. The metrics reported are on channel 1 source images.

| Row | Source Estimate | FCP Target | FCP | SDR (dB) | SI-SDR (dB) | PESQ | eSTOI |
|-----|-----------------|------------|-----|----------|-------------|------|-------|
| 1 | anechoic $\tilde{S}_k$ | mixture (ch-1) $X_1$ | $\text{FCP}(X_1, \tilde{S}_k)$ | 22.0 | 19.8 | 4.15 | 0.974 |
| 2 | early (ch-2) $S^{\text{early}}_{k,2}$ | mixture (ch-1) $X_1$ | $\text{FCP}(X_1, S^{\text{early}}_{k,2})$ | 16.5 | 13.2 | 3.91 | 0.943 |
| 3 | reverb (ch-2) $S_{k,2}$ | mixture (ch-1) $X_1$ | $\text{FCP}(X_1, S_{k,2})$ | 14.7 | 11.3 | 3.59 | 0.917 |
| 4 | anechoic $\tilde{S}_k$ | reverb (ch-1) $S_{k,1}$ | $\text{FCP}(S_{k,1}, \tilde{S}_k)$ | 34.9 | 33.3 | 4.45 | 0.997 |
| 5 | early (ch-2) $S^{\text{early}}_{k,2}$ | reverb (ch-1) $S_{k,1}$ | $\text{FCP}(S_{k,1}, S^{\text{early}}_{k,2})$ | 18.2 | 14.5 | 4.15 | 0.967 |
| 6 | reverb (ch-2) $S_{k,2}$ | reverb (ch-1) $S_{k,1}$ | $\text{FCP}(S_{k,1}, S_{k,2})$ | 16.0 | 12.1 | 3.84 | 0.941 |

First, observe the first three rows in Table 3, where the FCP target is set to be the channel 1 mixture. In rows 1, 2, and 3, the source estimate is set to be the anechoic source, the early reverberant source, and the reverberant source, respectively. We can see that when the source estimate is set to be the anechoic source, the FCP filtering performance is much better than when the source estimate is early reverberant or reverberant. To see why this is the case, remember that in our signal

model in Eq. 3 in Sec. 2, the relative RIR from one channel source image to another channel source image involves a inverse filtering operation, which may not exist. This also explains why in row 2a-2b in Table 1, using an anechoic speech prior performs better than using the reverberant speech prior.

Row 4,5,6 in Table 3 shows the case when the FCP target is the channel 1 source image. Note that in row 4, when the source estimate is set to groud-truth anechoic source, the metrics is almost perfect, showing the correctness of the signal model used in Sec. 2.

## H. 2 and 3 Channels Spatialized WSJ0-2Mix Evaluation Results

*Table 4.* Evaluation results for 2-channel Spatialized WSJ0-2Mix. Note that the microphone positions are random for this dataset. Top results are emphasized in top1 , top2 , and top3 . Methods denoted with ∗ means it is impractical.

| Row | Methods | Unsup. | Array Agnostic | Prior | IVA Init. | Ref. Guide. | SDR (dB) | SI-SDR (dB) | PESQ | eSTOI |
|---|---|---|---|---|---|---|---|---|---|---|
| 0 | Mixture | - | - | - | - | - | 0.2 | 0.0 | 1.81 | 0.545 |
| 1a | Spatial Cluster | ✓ | ✓ | - | - | - | 7.9 | 6.5 | 2.38 | 0.689 |
| 1b | IVA | ✓ | ✓ | Laplace | - | - | 9.3 | 8.1 | 2.52 | 0.725 |
| 1c | IVA | ✓ | ✓ | Gaussian | - | - | 10.9 | 9.8 | 2.68 | 0.770 |
| 1d | UNSSOR | ✓ | ✗ | - | - | - | 0.3 | -2.7 | 1.78 | 0.478 |
| 2a | ArrayDPS-A | ✓ | ✓ | Anechoic | ✓ | ✓ | 14.5±0.7 | 13.7±0.7 | 3.32±0.08 | 0.853±0.014 |
| 2b | ArrayDPS-D | ✓ | ✓ | Anechoic | ✓ | ✓ | 5.6±4.0 | 3.6±4.6 | 2.34±0.45 | 0.652±0.114 |
| 3a | ArrayDPS-A-Max* | ✓ | ✓ | Anechoic | ✓ | ✓ | 15.3 | 14.6 | 3.41 | 0.870 |
| 3b | ArrayDPS-D-Max* | ✓ | ✓ | Anechoic | ✗ | ✓ | 11.2 | 9.9 | 2.96 | 0.799 |
| 4a | ArrayDPS-A-ML | ✓ | ✓ | Anechoic | ✓ | ✓ | 15.1 | 14.3 | 3.39 | 0.865 |
| 4b | ArrayDPS-D-ML | ✓ | ✓ | Anechoic | ✗ | ✓ | 10.4 | 8.9 | 2.87 | 0.773 |
| 5a | TF-GridNet-Spatial | ✗ | (2-mics) | - | - | - | 15.2 | 14.5 | 3.63 | 0.888 |

*Table 5.* Evaluation results for 3-channel Spatialized WSJ0-2Mix. Note that the microphone positions are random for this dataset. Top results are emphasized in top1 , top2 , and top3 . Methods denoted with ∗ means it is impractical.

| Row | Methods | Unsup. | Array Agnostic | Prior | IVA Init. | Ref. Guide. | SDR (dB) | SI-SDR (dB) | PESQ | eSTOI |
|---|---|---|---|---|---|---|---|---|---|---|
| 0 | Mixture | - | - | - | - | - | 0.2 | 0.0 | 1.81 | 0.545 |
| 1a | Spatial Cluster | ✓ | ✓ | - | - | - | 9.1 | 7.8 | 2.53 | 0.735 |
| 1b | IVA | ✓ | ✓ | Laplace | - | - | 10.4 | 8.5 | 2.66 | 0.750 |
| 1c | IVA | ✓ | ✓ | Gaussian | - | - | 13.9 | 12.1 | 3.07 | 0.842 |
| 1d | UNSSOR | ✓ | ✗ | - | - | - | 1.7 | -2.4 | 1.94 | 0.519 |
| 2a | ArrayDPS-A | ✓ | ✓ | Anechoic | ✓ | ✓ | 15.7±0.6 | 15.0±0.7 | 3.44±0.07 | 0.872±0.012 |
| 2b | ArrayDPS-D | ✓ | ✓ | Anechoic | ✓ | ✓ | 6.3±4.3 | 4.3±4.9 | 2.41±0.49 | 0.668±0.119 |
| 3a | ArrayDPS-A-Max* | ✓ | ✓ | Anechoic | ✓ | ✓ | 16.4 | 15.7 | 3.52 | 0.886 |
| 3b | ArrayDPS-D-Max* | ✓ | ✓ | Anechoic | ✗ | ✓ | 12.3 | 11.0 | 3.07 | 0.808 |
| 4a | ArrayDPS-A-ML | ✓ | ✓ | Anechoic | ✓ | ✓ | 16.3 | 15.5 | 3.49 | 0.881 |
| 4b | ArrayDPS-D-ML | ✓ | ✓ | Anechoic | ✗ | ✓ | 11.5 | 10.0 | 2.99 | 0.796 |
| 5a | TF-GridNet-Spatial | ✗ | (3-mics) | - | - | - | 15.5 | 14.8 | 3.65 | 0.892 |

In addition to the 4-channel Spatialized WSJ0-2Mix dataset's result in Table 2, we also report 2-channel (first 2 mics) and 3-channel (first 3 mics) results on Spatialized WSJ0-2Mix dataset, in Table 4 and Table 5 respectively.

For the 2-channel case, we first note that in row **1d** of Table 4, the UNSSOR does not work properly. This is actually as expected because UNSSOR assumes an over-determined scenario (Wang & Watanabe, 2023), which means the number of microphones should be larger than the number of speakers. In contrast, our ArrayDPS is able to work properly. In addition, our method easily outperforms any other unsupervised methods. Finally, we compare row **5a** with row **2a** and row **4a**, we find that the supervised method is slightly better than ArrayDPS-A-ML in all metrics for the 2-channel case.

For the 3-channel results shown in Table 5, the result is very similar to the 4-channel results shown in Table 2. In short, ArrayDPS-A outperforms all other unsupervised methods, while UNSSOR does not work properly in this 3-channel setting. As mentioned in Sec. F, UNSSOR is very sensitive to ISMS loss weight $\gamma$, but we still cannot find a working weight after an extensive parameter search. Note that we ensure that our training is correct as it works for both 4-channel Spatialized WSJ0-2Mix and 3-channel SMS-WSJ datasets. One explanation is that the training dataset contains samples recorded by different 3-channel ad-hoc microphone arrays, which makes it more difficult for UNSSOR to converge. We further compare ArrayDPS (row **2a, 4a**) with the supervised baseline (row **5a**), where ArrayDPS-A's mean SDR and SI-SDR can outperform the supervised TF-GridNet-Spatial.

## I. 6-Channel SMS-WSJ Evaluation Results

In addition to the 3-channel SMS-WSJ result reported in Table 1, we also report a full 6-channel (all mics used) result for SMS-WSJ dataset. We first compare row **1(a-c)** with row **2a**, where our ArrayDPS-A easily outperforms spatial clustering and IVA. Comparing row **1d** with row **2d**, we can see that ArrayDPS-A's mean metric score outperforms UNSSOR by about 0.7 dB in SDR and SI-SDR, but performs a bit worse in terms of PESQ and eSTOI. However, our maximum likelihood version ArrayDPS-A-ML (row **4a**) easily outperforms UNSSOR in all metrics by a large margin. Although in 6-channel setting, ArrayDPS-A-ML is still better than any other unsupervised methods, the supervised TF-GridNet-SMS in row **5a** leads our ArrayDPS by a large margin. Future research will be conducted to reduce this performance gap.

*Table 6.* Evaluation results for 6-channel SMS-WSJ. Methods denoted with † are results from UNSSOR (Wang & Watanabe, 2023), and methods denoted with ∗ mean it is impractical. Note that SMS-WSJ only contains samples with a fixed microphone array. Top results are emphasized in `top1` , `top2` , and `top3` .

| Row | Methods | Unsup. | Array Agnostic | Prior | IVA Init. | Ref. Guide. | SDR (dB) | SI-SDR (dB) | PESQ | eSTOI |
|---|---|---|---|---|---|---|---|---|---|---|
| 0 | Mixture | - | - | - | - | - | 0.1 | 0.0 | 1.87 | 0.603 |
| 1a | Spatial Cluster† | ✓ | ✓ | - | - | - | 11.9 | 10.2 | 2.61 | 0.735 |
| 1b | IVA† | ✓ | ✓ | Laplace | - | - | 10.6 | 8.9 | 2.58 | 0.764 |
| 1c | IVA | ✓ | ✓ | Gaussian | - | - | 14.7 | 13.4 | 3.07 | 0.865 |
| 1d | UNSSOR† | ✓ | × | - | - | - | 15.7 | 14.7 | 3.47 | 0.884 |
| 2a | ArrayDPS-A | ✓ | ✓ | Anechoic | ✓ | ✓ | 16.3±1.2 | 15.4±1.3 | 3.45±0.12 | 0.873±0.019 |
| 2b | ArrayDPS-D | ✓ | ✓ | Anechoic | ✓ | ✓ | 8.9±4.9 | 7.2±5.4 | 2.67±0.54 | 0.743±0.115 |
| 3a | ArrayDPS-A-Max* | ✓ | ✓ | Anechoic | ✓ | ✓ | 17.6 | 16.9 | 3.59 | 0.896 |
| 3b | ArrayDPS-D-Max* | ✓ | ✓ | Anechoic | × | ✓ | 15.1 | 14.0 | 3.32 | 0.873 |
| 4a | ArrayDPS-A-ML | ✓ | ✓ | Anechoic | ✓ | ✓ | 17.4 | 16.6 | 3.56 | 0.891 |
| 4b | ArrayDPS-D-ML | ✓ | ✓ | Anechoic | × | ✓ | 14.6 | 13.5 | 3.27 | 0.862 |
| 5a | TF-GridNet-SMS† | × | × | - | - | - | 19.4 | 18.9 | 4.08 | 0.949 |

## J. 3-Speaker Evaluation Results

For 3-speaker source separation, we evaluate on the spatialized WSJ0-3Mix dataset, which is the 3-speaker version of the ad-hoc spatialized WSJ0-2Mix dataset. We evaluate on the first 4 microphones, where the results are shown in Table 7. As shown in Row **2a-5a**, ArrayDPS outperforms the supervised baseline by a large margin in all metrics. If we sample five samples and then take the maximum likelihood one (Row **4a**), ArrayDPS outperforms supervised TF-GridNet-Spatial by more than 3 dB in terms of SDR and SI-SDR. This shows the potential of ArrayDPS in more diverse acoustic scenarios.

*Table 7.* Evaluation results for 3-speaker reverberant speech separation on 4-channel Spatialized WSJ0-3Mix. Note that the microphone positions are random for this dataset. Top results are emphasized in top1 , top2 , and top3 . Methods denoted with ∗ means it is impractical.

| Row | Methods | Unsup. | Array Agnostic | Prior | IVA Init. | Ref. Guide. | SDR (dB) | SI-SDR (dB) | PESQ | eSTOI |
|---|---|---|---|---|---|---|---|---|---|---|
| 0 | Mixture | - | - | - | - | - | -3.0 | -3.3 | 1.50 | 0.371 |
| 1a | Spatial Cluster | ✓ | ✓ | - | - | - | 6.9 | 5.6 | 2.10 | 0.625 |
| 1b | IVA | ✓ | ✓ | Laplace | - | - | 5.9 | 3.8 | 2.10 | 0.574 |
| 1c | IVA | ✓ | ✓ | Gaussian | - | - | 9.6 | 7.6 | 2.46 | 0.701 |
| 1d | UNSSOR | ✓ | × | - | - | - | -2.8 | -6.1 | 1.50 | 0.313 |
| 2a | ArrayDPS-A | ✓ | ✓ | Anechoic | ✓ | ✓ | 12.8±1.2 | 11.8±1.3 | 3.11±0.16 | 0.791±0.028 |
| 3a | ArrayDPS-A-Max∗ | ✓ | ✓ | Anechoic | ✓ | ✓ | 14.3 | 13.4 | 3.31 | 0.816 |
| 4a | ArrayDPS-A-ML | ✓ | ✓ | Anechoic | ✓ | ✓ | 14.0 | 13.0 | 3.27 | 0.807 |
| 5a | TF-GridNet-Spatial | × | (4-mics) | - | - | - | 10.6 | 9.7 | 2.85 | 0.747 |

# K. Ablations on Sensitivity to Hyperparameters

To evaluate our ArrayDPS's sensitivity and robustness to hyper-parameters, we conduct ablation experiments on the likelihood score guidance $\xi$, the sampling starting noise level $\tau_{\max}$, FCP filter length $F$, the number of steps $N_{fg}$ to use the IVA initialized filters, whether to use Gaussian or Laplace prior for IVA initialization, and whether to use the IVA separated source as diffusion initialization (line 8 of Algorithm 1). For each parameter, we do ablations by modifying it starting from the default parameter mentioned in Appendix F. All the ablations are conducted on the first 50 samples of the SMS-WSJ validation set. Results are shown in Table 8-13 .

*Table 8.* Ablation study for $\xi$.

| $\xi$ | 1.2 | 1.4 | 1.6 | 1.8 | 2.0 | 2.2 | 2.4 | 2.6 | 2.8 |
|---|---|---|---|---|---|---|---|---|---|
| SI-SDR | 15.2±1.1 | 15.5±1.1 | 15.6±1.0 | 15.6±1.2 | 15.3±1.6 | 15.0±1.5 | 15.0±1.6 | 15.2±2.6 | 15.1±1.8 |

*Table 9.* Ablation study for $\tau_{\max}$.

| $\tau_{\max}$ | 0.5 | 0.6 | 0.7 | 0.8 | 0.9 | 1.0 | 1.1 |
|---|---|---|---|---|---|---|---|
| SI-SDR | 16.1±0.9 | 16.0±0.8 | 15.8±1.2 | 15.6±1.1 | 15.1±1.7 | 14.8±1.8 | 14.3±2.1 |

*Table 10.* Ablation study for FCP filter length $F$.

| $F$ | 10 | 11 | 12 | 13 | 14 | 15 | 16 |
|---|---|---|---|---|---|---|---|
| SI-SDR | 15.8±1.3 | 15.5±1.6 | 15.6±1.2 | 15.1±1.7 | 14.9±1.7 | 14.9±1.5 | 14.8±1.6 |

*Table 11.* Ablation study for filter guidance steps $N_{fg}$.

| $N_{fg}$ | 70 | 80 | 90 | 100 | 110 | 120 | 130 |
|---|---|---|---|---|---|---|---|
| SI-SDR | 15.0±1.7 | 15.2±1.7 | 14.6±2.1 | 15.2±1.7 | 15.4±1.2 | 15.5±1.1 | 15.5±1.1 |

*Table 12.* Ablation study for IVA prior used for IVA initialization.

| IVA Prior | Gaussian | Laplace |
|---|---|---|
| SI-SDR | 15.3±1.6 | 14.8±1.6 |

*Table 13.* Ablation study for using IVA separated sources as diffusion initialization (line 8 of Algorithm 1).

| Diffusion Initialization | Yes | No |
|---|---|---|
| SI-SDR | 15.3±1.6 | 13.3±2.3 |

## L. Filter and Source Visualization

Figure 3- 6 gives visualization of the separated sources and the final FCP estimated relative RIRs. In each figure of Figure 3- 6, the first row contains the ground-truth anechoic source 1 $\tilde{s}_1$, the ground-truth RIR $h_{1,1}$ (from anechoic source to reference channel), and the reference-channel reverberant source 1 $s_{1,1}$. Obviously, the third signal is the convolution of the first two signals. Then the second row shows the same for our ArrayDPS, i.e., separated virtual source 1 $\hat{s}_0$, the final FCP estimated relative RIR $\hat{g}_{0\rightarrow1}^1$, and the separated reference-channel source 1 $\hat{s}_{1,1}$. These two adjacent rows allow direct comparison between the ground-truth and ArrayDPS. Similarly, row 3 and row 4 show results for source 2.

These figures show us the difference between the final FCP estimated filter and the ground truth RIR. Those two signals might not be aligned, but sometimes show similar structure on the spectrogram. On the other hand, the virtual source separated by ArrayDPS is more like the anechoic signal rather than the reverberant signal. We explain this for two reasons: 1) the diffusion model is trained mostly on clean anechoic speech, hence the inclination to generate anechoic sources; 2) FCP performs much better when the input source signal is anechoic (see Appendix G). This means outputting an anechoic source (instead of a reverberant one) would satisfy a higher likelihood.

Our demo site also shows virtual sources with 2-speaker separation. We urge readers to listen to the virtual source samples and compare them with the final output and the ground-truth anechoic source. It appears that ArrayDPS is indeed accomplishing some dereverberation, however, we do not want to over-claim here and intend to verify this promise in future research.

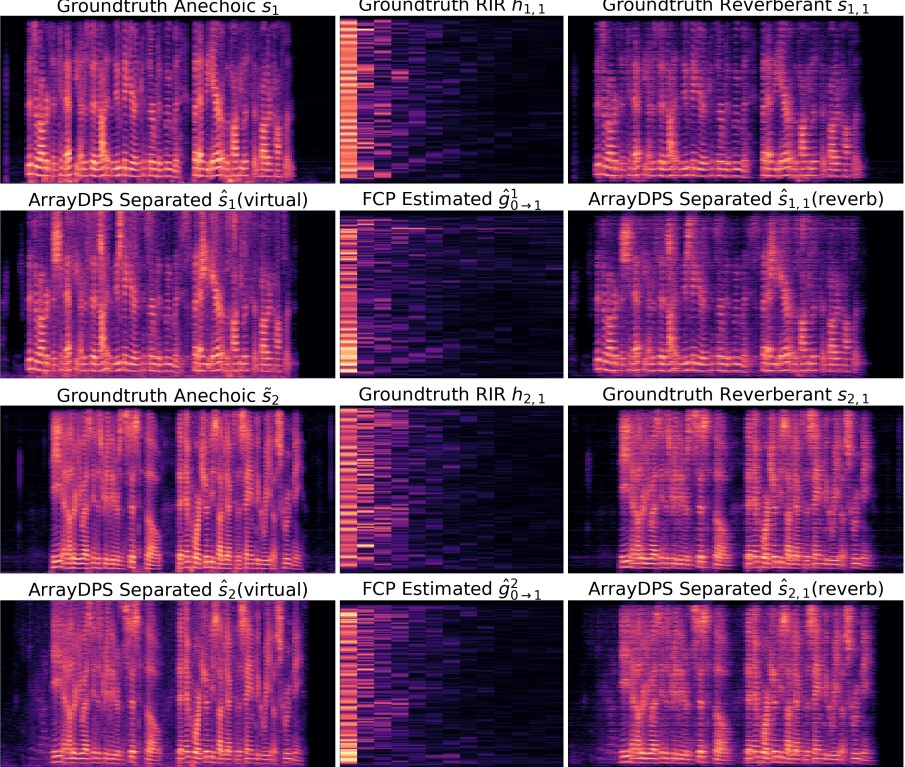

*Figure 3.* SMS-WSJ Sample 0_442c040o_443c040g Visualization

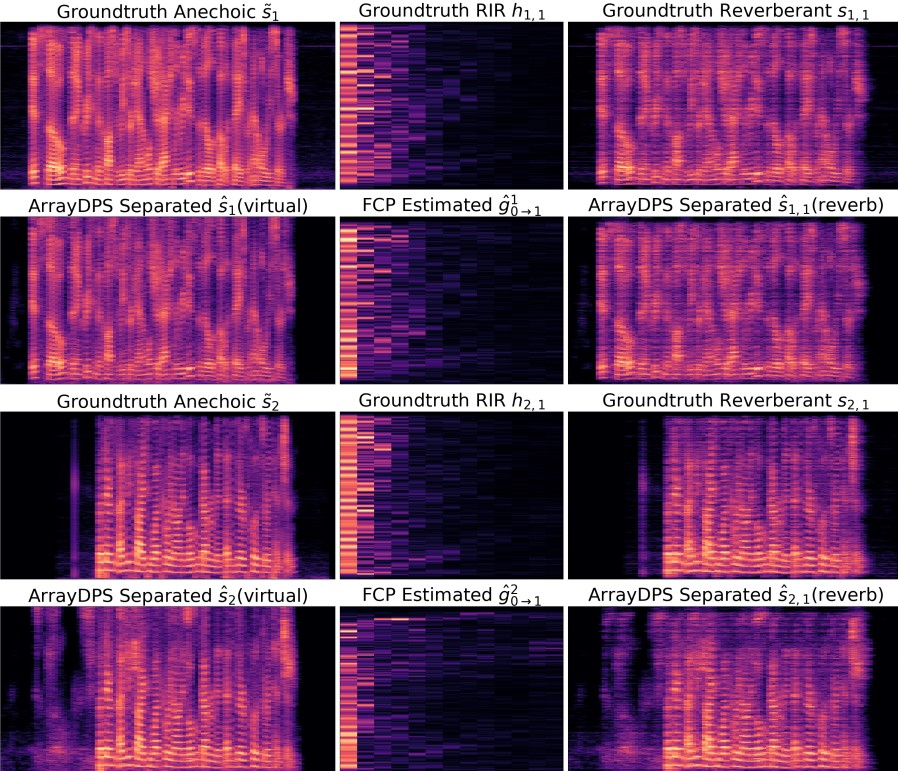

*Figure 4.* SMS-WSJ Sample 1015_446c0415_442c040c Visualization

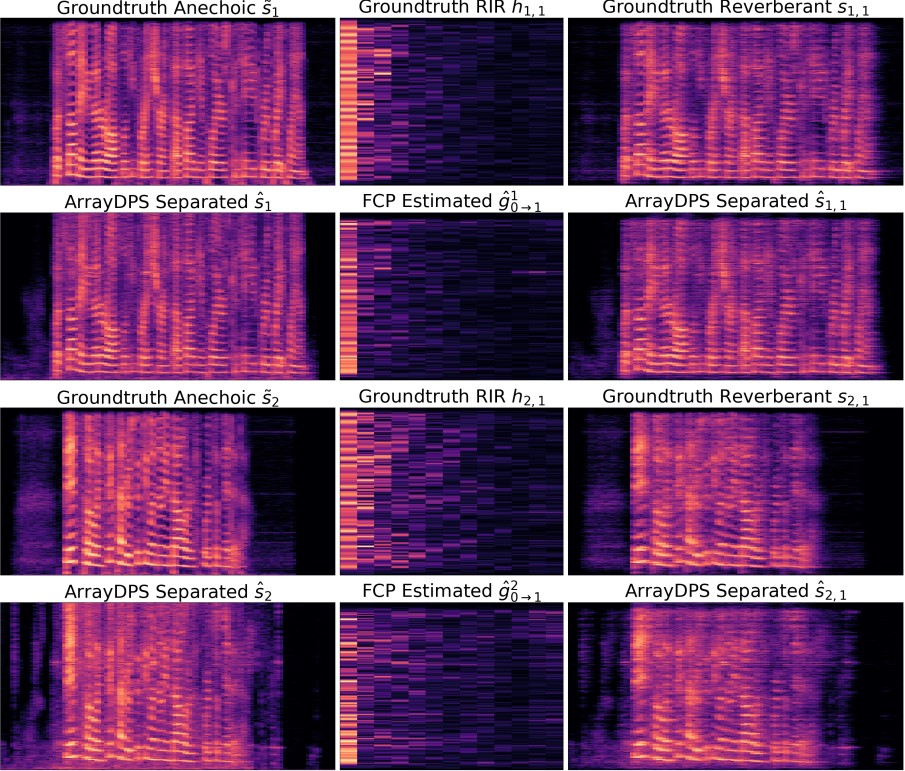

*Figure 5.* SMS-WSJ Sample 1120_445c040c_441c040m Visualization

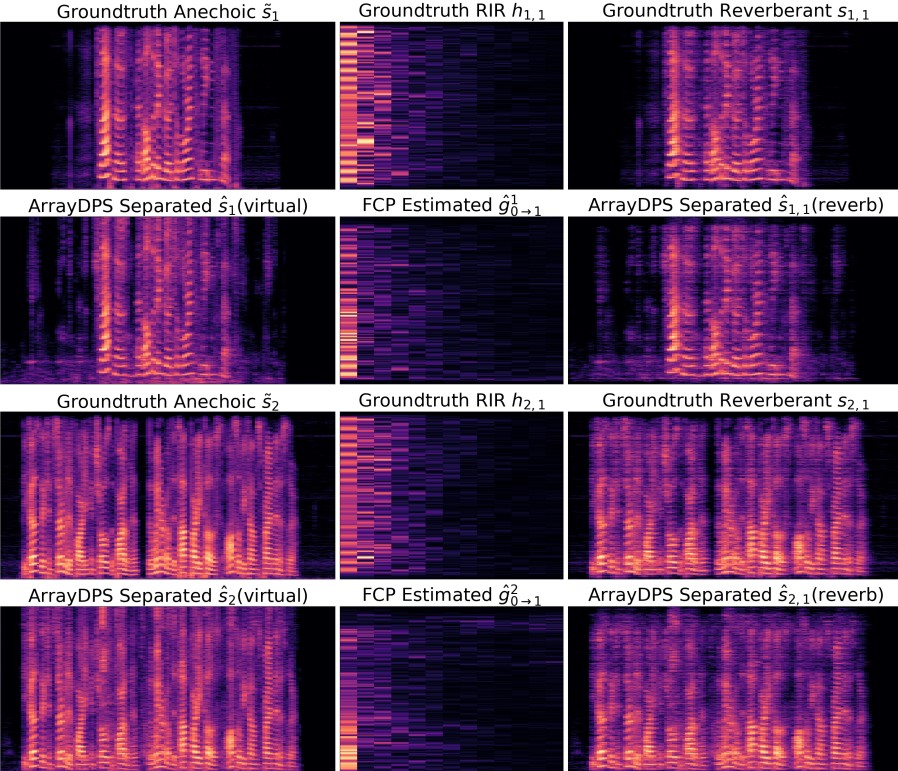

*Figure 6.* SMS-WSJ Sample 999_441c040c_447c040k Visualization

