# OpenReview forum: "ArrayDPS: Unsupervised Blind Speech Separation with a Diffusion Prior"
_ICML.cc/2025/Conference — ICML 2025 poster_

### Official Review · Reviewer_WSKB · 2025-03-08

**Overall Recommendation:** 4

**Summary:**

This paper presents a diffusion based method for speech separation using microphone arrays. The core idea is to use a DPS based approach where they alternative between diffusion steps, and an estimation of the reverberation/mixing matrix A. The matrix A is necessary to calculate the conditional likelihood updates required by DPS. Results are presented on WSJ synthetic mixtures of audio on different array configurations.

**Claims And Evidence:**

The authors do a good job of presenting evidence to back their claims. Experiment results with numerous ablation studies are presented, and there are qualitative audio examples that let the listener judge for themselves. The show that different array configurations can be used at inference time.

**Essential References Not Discussed:**

N/A

**Experimental Designs Or Analyses:**

The experiments are good. The authors evaluate many metrics (SDR, SI-SDR, PESQ, eSTOI) which is nice, and also the WSJ mix is standard. The ablation studies are extremely thorough as well. As I mentioned earlier, I would like to see some real-world examples and examples with more than 2 speakers.

**Methods And Evaluation Criteria:**

The authors compare against numerous baselines like TF Gridnet. There are other baselines that could be included (sepformer, demucs, conv tasnet) but since TF Gridnet is state of the art, it suffices. The dataset used, WSJ is also standard and a good choice for comparison.

I think the method is very good, but the authors could have done a lot more to strengthen the evaluation. Firstly, a major advantage of this generative approach is that it can handle any number of speakers during inference. But the authors only test with 2 speakers from the WSJ mix, never showing a result with more speakers. Also, I would like to see some real results captured on real-microphone data. All the WSJ mixtures are synthetic, and I am skeptical whether the linear approximation A will hold up extremely well in real-world scenarios where there may be other influences on the captured sound. Given that the authors are pitching the value of the paper in its generality to different reverbs, mic array geometries etc, I would really like to see at least one real-world, non synthetic example captured on any kind of microphone array.

**Other Comments Or Suggestions:**

Overall I like the paper, as it feels like it tackles the blind source separation with DPS in an elegant way. I think the main novelty is in the estimation of the A matrix to be used in DPS (otherwise it's just DPS applied to a linear mixture of sources). This could be conveyed a bit better. Also since that is a big part, it would be nice to see how well the predicted A matrix aligns with some ground truths in synthetically rendered scenarios.

**Other Strengths And Weaknesses:**

The writing is clear. I do wonder if algorithm 1 and 2 can be simplified to distill the essence of the key high level meaning with more details left in the appendix. Also figure 1 has a lot going on. I don't think all the formulas are needed in the figure, instead the high level ideas should be sufficient and make it easier to follow.

**Questions For Authors:**

Have you tried the algorithm with more than 2 speakers? At how many speakers does it fail?

**Relation To Broader Scientific Literature:**

This paper does a good job of tying in DPS, a recent and innovative diffusion work, with the problem of multi-channel speech separation. It does a good job of bridging the gap between traditional multi-mic source separation and newer diffusion methods.

**Theoretical Claims:**

The overall theory behind the likelihood estimation and IVA seems sound. Also estimating A in the frequency domain makes a lot of sense.

---

> ### Author Rebuttal · Authors · 2025-03-31
>
> We thank the reviewer for the thoughtful and encouraging comments.
> >Q: I think the method is very good, but the authors could have done a lot more to strengthen the evaluation. The authors only test with 2 speakers from the WSJ mix, never showing a result with more speakers.
>
> We sincerely appreciate the reviewer’s kind remarks and will work hard to strengthen the evaluation.
> During the rebuttal phase, we ran new experiments on a commonly used 3-speaker reverberant source separation dataset.
> The results are in our rebuttal to Reviewer Go2n, where we show that ArrayDPS performs surprisingly well. Demos for the 3-speaker separation case are also updated in the demo page **https://arraydps.github.io/ArrayDPSDemo/**.
>
> >Q: Also, I would like to see some real results captured on real-microphone data. All the WSJ mixtures are synthetic, and I am skeptical whether the linear approximation A will hold up extremely well in real-world scenarios where there may be other influences on the captured sound. I would really like to see at least one real-world, non-synthetic example captured on any kind of microphone array.
>
> To validate our method in real-world conditions, we recorded 15 mixtures in a 7m x 4m x 2.7m room. Two volunteers speak simultaneously under weak environmental noise and the mixtures are recorded by 3 microphones. Demos and recordings are in https://github.com/ArrayDPS/ArrayDPSDemo/tree/main/demo_results_realworld and are updated in the demo site **https://arraydps.github.io/ArrayDPSDemo/**. These new real world results sound consistent with the earlier synthetic results reported in the paper.
>
> >Q: The writing is clear. I do wonder if algorithms 1 and 2 can be simplified to distill the essence of the key high level meaning with more details left in the appendix. Also figure 1 has a lot going on. I don't think all the formulas are needed in the figure, instead, the high level ideas should be sufficient and make it easier to follow.
>
> Thanks for your suggestions! Reviewer Go2n also suggests simplifying Algorithm 1 by abstracting out the second-order Heun sampler in EDM [2]. We welcome the advice. For the figures, we will remove some unnecessary formulas to make it clearer.
>
> >Q: Overall, I like the paper, as it feels like it tackles the blind source separation with DPS in an elegant way. I think the main novelty is in the estimation of the A matrix to be used in DPS (otherwise it's just DPS applied to a linear mixture of sources). This could be conveyed a bit better. Also, since that is a big part, it would be nice to see how well the predicted A matrix aligns with some ground truths in synthetically rendered scenarios.
>
> We are grateful for the reviewer’s encouraging comments and are excited to answer the question related to the predicted A matrix. In the demo page **https://arraydps.github.io/ArrayDPSDemo/**,  we updated a section called **Filter and Source Visualization**, where we show 6 figures corresponding to 6 mixtures.
>
> In each figure, the first row contains source 1 anechoic source signal $\tilde{s_1}$, the ground-truth STFT domain RIR filter $h_{1,1}$ (from anechoic source to reference channel), and the reference-channel reverberant source 1 $s_{1,1}$. Of course, the third signal is the convolution of the first two signals.
> Then the second row shows the same for ArrayDPS, i.e., separated virtual source 1 $\hat{s_1}$, the final FCP estimated filter $\hat{g}^1_{0\rightarrow1}$, and the final reference-channel source 1 output $\hat{s_{1,1}}$.  These two adjacent rows allow direct comparison between ground truth and ArrayDPS.
> Similarly, row 3 and row 4 show results for source 2.
>
> We can visually see the difference between the final FCP estimated filter and the ground truth RIR through these figures. Those two signals might not be aligned, but sometimes show similar structure on the spectrogram. On the other hand, the virtual source separated by ArrayDPS is more like the anechoic signal rather than the reverberant signal. We explain this with two reasons:
> First, the diffusion model is trained on mostly anechoic clean speech, hence the inclination to generate anechoic sources.
> Second, FCP performs much better when the input source signal is anechoic (see Appendix G, and our rebuttal to reviewer Go2n, about why we are using the virtual source model). This means outputting an anechoic source (instead of a reverberant one) would satisfy a higher likelihood.
>
> We also updated our demo site’s 2 speakers separation section with virtual sources. We urge readers to listen to the virtual source samples and compare them with the final output and the ground-truth anechoic source. It appears that ArrayDPS is indeed accomplishing some dereverberation, however, we do not want to over-claim here and intend to verify this promise in future research.
>
> [1] Karras, Tero, et al. “Elucidating the design space of diffusion-based generative models.” Advances in neural information processing systems 35 (2022): 26565-26577.

---

### Official Review · Reviewer_L5PQ · 2025-03-12

**Overall Recommendation:** 4

**Summary:**

This paper proposes ArrayDPS for multi-channel blind speech separation. The main contributions claimed by the authors are as follows:
1. Proposal of an array-agnostic unsupervised learning-based blind speech separation model that is robust to microphone array variations.
2. Introduction of the first method utilizing Diffusion Posterior Sampling (DPS) to solve the multi-channel array inverse problem.
3. Achieving performance comparable to supervised methods while surpassing existing unsupervised approaches in the field of speech separation.

**Claims And Evidence:**

Strengths
1. The limitations of conventional supervised learning-based approaches using synthetic dataset are well-identified, and the formulation of blind speech separation as an inverse problem is clearly structured.
2. The overall overview of the proposed ArrayDPS using DPS is well summarized, and both the detailed explanation and derivation of the posterior score approximation using the prior score and likelihood score are clearly presented.
3. As an unsupervised and array-agnostic approach in the field of speech separation, ArrayDPS surpasses existing limitations and demonstrates its potential applicability to real-world applications.

Weaknesses
1. Limitations in the experiments conducted to support the proposed claims. (detailed in the “Experimental Designs or Analyses”)
2. Minor issue related to inconsistencies in mathematical formulation and function usage. (detailed in the “Theoretical Claims”)
3. Minor issues related to the paper structure and typos. (detailed in “Other Comments or Suggestions”)

**Essential References Not Discussed:**

As mentioned in “Experimental Designs Or Analyses”, it is necessary to consider existing array-agnostic specialized models, such as VarArray [1].

[1] Yoshioka, Takuya, et al. "VarArray: Array-geometry-agnostic continuous speech separation." ICASSP 2022-2022 IEEE International Conference on Acoustics, Speech and Signal Processing (ICASSP). IEEE, 2022.

**Experimental Designs Or Analyses:**

Strengths
1. The distinction between "unsupervised" and "array-agnostic" models in the Tables enhances the clarity and intuitiveness of the results.
2. The robustness of the proposed method to microphone array variations is well-supported through experiments conducted on random microphone arrays (Spatialized WSJ0-2mix dataset).

Weaknesses
1. The authors point out the generalizability issue from supervised learning-based approaches using synthetic dataset. However, the dataset used in this paper also relies on simulated RIRs, raising the same generalization concerns.
2. The "Max" model selects the best of five samples for each test set, which may not be a fair comparison. It would be helpful to explicitly state in the Table whether this approach is “practical”, and consider refraining from using it for ranking in the top results to ensure a fair evaluation.
3. The array-agnostic baseline model used is relatively outdated, and TF-GridNet-Spatial is merely a supervised model trained on diverse microphone arrays. Comparison with a more relevant array-geometry-robust model, such as VarArray [1], should be considered.
4. The impact of IVA initialization on performance is significant, yet there is a lack of detailed analysis regarding its contribution.

[1] Yoshioka, Takuya, et al. "VarArray: Array-geometry-agnostic continuous speech separation." ICASSP 2022-2022 IEEE International Conference on Acoustics, Speech and Signal Processing (ICASSP). IEEE, 2022.

**Methods And Evaluation Criteria:**

The evaluation metrics used for blind speech separation—SDR, SI-SDR, PESQ, and eSTOI—seem appropriate and reasonable for assessing model performance.

**Other Comments Or Suggestions:**

1. Reference formatting issues:
 - The citation format for ICASSP conference papers is inconsistent.
 - Capitalization in the titles of references should be corrected.
2. Paper structure improvement:
 - The "Related Work" section (Section 5) should be placed earlier (before the “Problem Formulation”) in the paper for a more natural flow.
3. Typos:
 - In the Appendix B.1, the sentence starting with "First, ~" contains an extra closing parenthesis ")".
 - "row1,2,3" → "rows 1, 2, and 3"
 - "Table G" → "Table 3"
 - “Algorithm 3.5” → “Sec. 3.5”

**Other Strengths And Weaknesses:**

As mentioned above, the problem formulation and the proposed solution are clearly presented. However, the experimental validation and analysis are insufficient to fully support the proposed claims.

**Questions For Authors:**

I have no any questions for the authors.

**Relation To Broader Scientific Literature:**

1. Unlike conventional methods that rely on fixed microphone arrays, the proposed array-agnostic speech separation method offers greater flexibility for real-world applications.
2. The performance improvement in unsupervised approaches suggests potential applications in other unsupervised learning domains.

**Theoretical Claims:**

The process of estimating the likelihood score from the log likelihood approximation and the prior score from the diffusion denoising model to compute the posterior score in the ODE is logically well-explained. Furthermore, the Appendix clearly presents the derivation, analysis and validation of the posterior score approximation.

However, in Algorithm 1, the function GET_SCORE(.) calls Algorithm 2, but the input used in Algorithm 1 (line 20) differs from the input specified in Algorithm 2, making its usage unclear.

---

> ### Author Rebuttal · Authors · 2025-03-31
>
> We thank the reviewer for the detailed and constructive comments. New results from real-world experiments and a 3-speaker dataset are discussed in the rebuttal to reviewer Go2n. Corresponding demos are updated in **https://arraydps.github.io/ArrayDPSDemo/**.
>
> >Q: GET_SCORE(.) in Algorithm 2 contains wrong arguments.
>
> This is a typo, where $D_\theta$ should be removed and $\hat{G}^{k, \text{IVA}}_{1\rightarrow c}$ should be inside the parentheses. We are sincerely sorry and will fix it.
>
> >Q: The dataset relies on simulated RIRs, raising generalization concerns.
>
> To validate our method in real-world conditions, we recorded 15 mixtures in a 7m x 4m x 2.7m room. Two volunteers speak simultaneously with weak environmental noise and are recorded by 3 microphones. Demos and recordings are in https://github.com/ArrayDPS/ArrayDPSDemo/tree/main/demo_results_realworld and are updated in the demo site.
>
> >Q: The "Max" model is impractical and not fair for comparison.
>
> Thanks for pointing this out. We will remove it from the ranking highlight and mark those as impractical.
>
> >Q: The array-agnostic baseline model used is relatively outdated, and TF-GridNet-Spatial is merely a supervised model trained on diverse microphone arrays. Comparison with a more relevant array-geometry-robust model, such as VarArray [1], should be considered.
>
> We understand the concern that TF-GridNet [2], although a very strong baseline, is not designed in an array-agnostic manner. However, we do not think this TASLP 2023 SOTA paper is outdated due to its inability to handle any number of microphone inputs. Training on 3-channel ad-hoc arrays could result in better performance when evaluated on the same 3-channel scenarios. Of course, this needs further research for validation as there is no existing array-agnostic version of TF-GridNet for direct comparison.
>
> For the VarArray baseline, we are a bit unsure about it as: (1) it follows FaSNet-TAC [3], but only reports word error rate and does not compare with FaSNet-TAC, (2) recent work [4] shows that it performs worse than FaSNet-TAC, and (3) to the best of our knowledge, there is no open-source implementation.
>
> In light of this, we directly trained an array-agnostic FaSNet-TAC on WSJ0-2Mix with a variable number of channels, and report the separation results below:
> |FaSNet-TAC|mics|SDR|SI-SDR|PESQ|ESTOI|
> |-|-|-|-|-|-|
> |Spatialized-WSJ|2|8.2|7.4|2.26|0.663|
> |Spatialized-WSJ|3|9.5|8.6|2.37|0.689|
> |Spatialized-WSJ|4|9.9|9.0|2.41|0.700|
> |SMS-WSJ|3|9.3|8.5|2.29|0.704|
> |SMS-WSJ|6|9.5|8.7|2.33|0.711|
>
> We understand that this is by no means a fair baseline as it's using a 2020 backbone network. We will further look into a TAC version of TF-GridNet to fully clarify this issue.
>
> >Q: The impact of IVA initialization on performance is significant but lacks analysis.
>
> Reviewer KMQX also mentioned this, so we refer to our rebuttal to KMQX, where we report three ablations around the IVA initialization. We first compare  ArrayDPS initialized by IVA-Gaussian and IVA-Laplace. The second ablation is for parameter $N_{fg}$ (Algo. 2 line 9), which is the initial number of steps to use the IVA initialized filter. The third ablation compares using warm initialization or not in Algorithm 1, line 8. We believe these are the most important parameters to analyze, as the first shows ArrayDPS's sensitivity to IVA performance (Laplace prior performs worse), the second shows at what sampling time we should stop using the initialization and let FCP learn the filter, and the last one shows the effectiveness of initializing the diffusion starting point with IVA.
>
> >Q: Reference formatting issues
>
> Thanks! We realized there are inconsistencies and issues in the references. We will fix them.
>
> >Q: Related work should be placed earlier.
>
> Thanks for the suggestion! We thought about this but made the call to move it to the end primarily because it would contain background material from multiple domains, including unsupervised separation, diffusion posterior sampling, and RIR estimation, etc. We thought a long related work may interrupt the flow of the paper.
>
> >Q: line 739 has an extra ")", "row1,2,3" → "rows 1, 2, and 3", "Table G" → "Table 3", "Algorithm 3.5" → "Sec. 3.5"
>
> Apologies. We will fix all these typos.
>
> [1] Yoshioka, Takuya, et al. "VarArray: Array-geometry-agnostic continuous speech separation." ICASSP 2022-2022 IEEE International Conference on Acoustics, Speech and Signal Processing (ICASSP). IEEE, 2022.
>
> [2] Z.-Q. Wang, et al. "TF-GridNet: Integrating Full- and Sub-Band Modeling for Speech Separation," IEEE/ACM TASLP, vol. 31, pp. 3221–3236, 2023.
>
> [3] Luo, Yi, et al. "End-to-end microphone permutation and number invariant multi-channel speech separation." ICASSP 2020-2020 IEEE international conference on acoustics, speech and signal processing (ICASSP). IEEE, 2020.
>
> [4] Chen, Weiguang, et al. UniArray: Unified Spectral-Spatial Modeling for Array-Geometry-Agnostic Speech Separation. arXiv preprint arXiv:2503.05110, 2025.

---

> > ### Comment · Reviewer_L5PQ · 2025-04-04
> >
> > Thank you for your response to our questions. Most of our concerns have been addressed, and we intend to raise our rating. However, as mentioned in your rebuttal to Reviewer KMQX, we hope that the analysis regarding initialization will be clearly reflected.

---

> > > ### Author Response · Authors · 2025-04-04
> > >
> > > Thank you for your positive feedback and for raising your rating. We're glad that most of your concerns have been addressed. As noted, we will ensure that the final version of the paper, if accepted, includes more solid and detailed analysis on IVA initialization parameters, along with other important parameters we mentioned in rebuttal to KMQX.
> > >
> > > We also update our results of a much more recent array-agnosic baseline called USES [1]. We train this model on the variable-number-of-channel Spatialized WSJ dataset (same as the FaSNet-TAC baseline we mentioned before). Below shows our results on all 2-speaker datasets in the paper.
> > >
> > > | USES       | mics| SDR  | SI-SDR | PESQ | ESTOI |
> > > |------------|------------|------|--------|------|-------|
> > > | Spatialized-WSJ| 2          | 12.4 | 11.7   | 3.29 | 0.839 |
> > > | Spatialized-WSJ| 3          | 12.9 | 12.2   | 3.36 | 0.850 |
> > > | Spatialized-WSJ| 4          | 13.1 | 12.4   | 3.39 | 0.854 |
> > > | SMS-WSJ| 3          | 11.3 | 10.6   | 3.01 | 0.818 |
> > > | SMS-WSJ| 6          | 11.3 | 10.7   | 3.03 | 0.820 |
> > >
> > > [1] W. Zhang, K. Saijo, Z. -Q. Wang, S. Watanabe and Y. Qian, "Toward Universal Speech Enhancement For Diverse Input Conditions," 2023 IEEE Automatic Speech Recognition and Understanding Workshop (ASRU), Taipei, Taiwan, 2023, pp. 1-6, doi: 10.1109/ASRU57964.2023.10389733.

---

### Official Review · Reviewer_KMQX · 2025-03-13

**Overall Recommendation:** 3

**Summary:**

The paper introduces ArrayDPS, an unsupervised, generative, and array-agnostic method for blind speech separation (BSS). ArrayDPS leverages a diffusion-based generative model as a speech prior, combined with a novel approximation method for the intractable likelihood arising due to unknown array geometries and unknown room impulse responses (RIR). The key conceptual innovation is employing Diffusion Posterior Sampling (DPS) combined with the Forward Convolutive Prediction (FCP) algorithm, allowing estimation of relative room impulse responses (RIRs). Notably, ArrayDPS achieves competitive performance with supervised approaches while significantly surpassing previous unsupervised techniques. Experiments are performed on standard benchmarks, SMS-WSJ (fixed array) and Spatialized WSJ0-2Mix (ad-hoc array scenarios), showcasing ArrayDPS’s generalization across different microphone arrays and acoustics. The paper demonstrates a notable advancement in unsupervised multi-channel speech separation by integrating strong priors learned from single-channel diffusion models with likelihood approximations tailored for blind array conditions.

**Claims And Evidence:**

The primary claim is that ArrayDPS achieves competitive performance compared to supervised methods and outperforms unsupervised methods significantly. This claim is reasonably supported by comprehensive experiments on recognized benchmark datasets (SMS-WSJ, Spatialized WSJ0-2Mix). However, the paper could strengthen its evidence by evaluating more diverse real-world scenarios or noisy acoustic conditions to demonstrate true generalizability. Additionally, the reproducibility of the likelihood approximation and posterior sampling processes should be validated more thoroughly with sensitivity analysis or additional controlled experiments.

**Essential References Not Discussed:**

The paper provides an extensive reference list.

**Experimental Designs Or Analyses:**

The experimental design is strong and robust. It clearly explained experiments and configurations are detailed thoroughly, and extensive ablation studies (e.g., the effect of IVA initialization, anechoic vs reverberant priors, and reference-channel guidance) are performed. Also, the empirical approximation assumptions are explicitly analyzed and justified.

**Methods And Evaluation Criteria:**

**Proposed Methods**
- The paper applies an existing DPS framework but modifies the likelihood term to handle multi-channel signals. The authors approximate the “unknown” part of the likelihood with a linear filter estimation approach (Forward Convolutive Prediction), thereby modeling room acoustics implicitly.
- They combine a generic scoring from a single-speaker diffusion model with a relative RIR fitting routine in each sampling step. This step yields a multi-channel separation pipeline that does not require supervised training on a matched array geometry.

**Evaluation Criteria**
- The paper focuses on standard separation metrics: SDR, SI-SDR, PESQ, and eSTOI. They compare the mean (and also best sample) across multiple runs, which is helpful in understanding both typical performance and best-case scenario.
- They use well-established corpora (SMS-WSJ, Spatialized WSJ0-2Mix) that test both fixed and ad-hoc array setups, thus covering a broad range of real-world scenarios.

Overall, the approach and benchmarks are appropriate. The reliance on standard speech separation metrics is conventional, though it might have been beneficial to analyze more robust real-room data or consider time/frequency domain direct speech intelligibility measures. The methodology is consistent with standard practices in speech separation research.

**Other Comments Or Suggestions:**

Please refer to the previous sections.

**Other Strengths And Weaknesses:**

**Strengths**

- ArrayDPS demonstrates notable originality in applying generative diffusion priors to a challenging unsupervised multi-channel blind separation scenario.
- The proposed framework elegantly avoids needing explicit microphone geometry knowledge, significantly improving usability and practical applicability to diverse hardware configurations.

**Weaknesses**

- The algorithm heavily relies on IVA initialization for stability and performance. This dependency somewhat reduces the claimed elegance and universality of the method.
- The standard deviation of the results (around ±1.2 dB) indicates that the method still exhibits notable instability, potentially limiting practical reliability.

**Questions For Authors:**

- Have you considered evaluating ArrayDPS on realistic datasets involving more complex acoustic conditions (real recordings, background noise, and dynamic speakers)? Could this impact your algorithm's stability and robustness?

- How sensitive is ArrayDPS to variations in hyperparameters such as diffusion schedule, IVA parameters, and RIR estimation settings?

- Is the computational complexity of ArrayDPS practical for real-time deployment scenarios? If not, could you briefly describe possible strategies for optimization or simplification?

**Relation To Broader Scientific Literature:**

ArrayDPS connects clearly with broader literature by **Extending Diffusion Posterior Sampling (DPS)**, previously employed in single-channel or supervised multi-channel scenarios, to unsupervised and array-agnostic BSS. It also provides a bridge from classic spatial clustering, ICA, IVA methods, and recent neural generative approaches (e.g., diffusion-based generative models).

**Theoretical Claims:**

The paper primarily provides empirical contributions without heavy theoretical claims. However, the authors provide a theoretical justification for the equivalence of FCP to maximum likelihood estimation in the supplementary derivation (Theorem B.1). The derivations in Appendix B are clearly stated and appear mathematically correct and consistent.

---

> ### Author Rebuttal · Authors · 2025-03-30
>
> We thank the reviewer for the detailed and constructive comments.
> >Q: The paper could strengthen its evidence by evaluating more diverse real-world scenarios or noisy acoustic conditions to demonstrate true generalizability. Have you considered evaluating ArrayDPS on realistic datasets involving more complex acoustic conditions (real recordings, background noise, and dynamic speakers)? Could this impact your algorithm's stability and robustness?
>
> Thank you for the suggestion. We ran new experiments for the following scenarios:
> (1) Real-world mixture recordings in a 7m x 4m x 2.7m room, where 2 speakers speak simultaneously. The recordings are performed using 3 microphones and under weak background noise.
> (2) A 3-speaker dataset to test the separation performance of all 3 speakers.
> The real-world experiment maintains strong separation performance, while in the 3-speaker (synthetic dataset) experiment, ArrayDPS outperforms the baselines by a large margin (to our surprise). The main results are reported here: **https://arraydps.github.io/ArrayDPSDemo/**, and additional demos and recording files are available here: **https://github.com/ArrayDPS/ArrayDPSDemo/tree/main/demo_results_realworld**.
>
> >Q: Sensitivity to hyperparameters like diffusion schedule, IVA parameters, and RIR estimation settings?
>
> Thanks for raising this question. We ran some new ablations for six important parameters, namely:
> (1) the diffusion scheduler’s starting noise level $\tau_{\max}$,
> (2) IVA filter guidance steps $N_{fg}$ in Algo.2 line 9,
> (3) likelihood guidance $\xi$ in Eq. 57,
> (4) causal FCP filter length $F$,
> (5) initialized with IVA-Gaussian or IVA-Laplace (row 1b and 1c in Table 1), and
> (6) whether the warm initialization in line 8 of Algorithm 1 is used.
> Since inference is expensive, we show SI-SDR only for the first 30 mixtures of the 3-channel SMS-WSJ’s validation set (we sample each mixture 5 times). The default parameters are $\tau_{\max}=0.8, N_{fg}=100, F=13, \xi=2$ and IVA-Gaussian and warm initialization are used.
> For results below, we change one parameter at a time.
>
> |$\xi$|1.2|1.4|1.6|1.8| 2.0|2.2|2.4|2.6|2.8|
> |-|-|-|-|-|-|-|-|-|-|
> |SI-SDR|15.2±1.1|15.5±1.1|15.6±1.0|15.6±1.2|15.3±1.6|15.0±1.5|15.0±1.6|15.2±2.6|15.1±1.8|
>
> |$\tau_{max}$|0.5|0.6|0.7|0.8|0.9|1.0|1.1|
> |-|-|-|-|-|-|-|-|
> |SI-SDR|16.1±0.9|16.1±0.8|15.6±1.3|15.3±1.6|15.1±1.7|14.8±1.8|14.3±2.1|
>
> |$F$|10|11|12|13|14|15|16|
> |-|-|-|-|-|-|-|-|
> |SI-SDR|15.8±1.3|15.5±1.6|15.6±1.5|15.3±1.6|14.9±1.7|14.9±1.5|14.8±1.6|
>
> |$N_{fg}$|70|80|90|100|110|120|130|
> |-|-|-|-|-|-|-|-|
> |SI-SDR|15.0±1.7|15.2±1.7|14.6±2.1|15.3±1.6|15.2±1.7|15.4±1.2|15.5±1.1|
>
> |Initialization used|IVA-Gaussian|IVA-Laplace|
> |-|-|-|
> |SI-SDR|15.3±1.6|14.8±1.6|
>
> |Use Warm Initialization|Yes|No|
> |-|-|-|
> |SI-SDR|15.3±1.6|13.3+2.3|
>
> We believe the proposed algorithm is robust to most parameters, although the result is only on a 30-mixture subset. To make this more conclusive, we will finish the ablation on a larger dataset.
>
> >Q: Computational complexity for real-time deployment scenarios and possible strategies for optimization.
>
> Currently, ArrayDPS and all neural baselines in this paper are impractical for real-time deployment scenarios due to the non-causal processing and high computation requirements. However, we think there are opportunities to mitigate this hurdle. First, we can use diffusion distillation techniques to reduce sampling steps or even make it one step. Also, our algorithm can easily fit into the flow matching’s sampling ODE, which would be more efficient. To enable real-time processing, we need to modify the diffusion U-Net to be a causal architecture, and then update FCP and the likelihood calculation with an adaptive version; we expect this to be reasonably straightforward.
>
> >Q: The algorithm heavily relies on IVA initialization for stability and performance. This dependency somewhat reduces the claimed elegance and universality of the method.
>
> We agree with this. Currently, ArrayDPS is sensitive to initialization, bearing similarity to the Expectation Maximization (EM) algorithm. We will add a section to discuss this and also give insights for future directions (1. Inducing a filter prior during sampling. 2. In early steps, maintain some uncertainty for the filter instead of greedily optimizing using FCP).
>
> >Q: The result STD (around ±1.2 dB) exhibits notable instability, potentially limiting reliability.
>
> We respectfully disagree with this argument since our method is generative, which means that our method samples multiple plausible solutions as separation results. These solutions might differ at the sample level, but all sound plausible. We believe (±1.2dB) is a reasonable range for 5 samples (please verify this in our demo about how all 5 different samples sound correct, but have ±1.2dB fluctuation). However, without IVA initialization, ArrayDPS-D exhibits a fluctuation of ±4.6 dB, implying our algorithm lacks consistency without IVA.
>
> Thanks again for your insightful observations.

---

### Official Review · Reviewer_Go2n · 2025-03-14

**Overall Recommendation:** 3

**Summary:**

The paper proposes a method for blind source separation in scenarios where the microphone array and RIR are unknown. The authors adopt a virtual source model, mapping all sources to an imaginary reference microphone.
They formulate the separation problem as an inverse problem, where: (i) a single-source diffusion model serves as the prior; (ii) The unknown likelihood, which involves the RIR, is estimated in the frequency domain.
The method is unsupervised and relies on diffusion posterior sampling.
Finally, the authors compare their approach against both supervised and unsupervised methods.

**Claims And Evidence:**

- The authors present the virtual source model as a flexible framework, but its utility remains unclear throughout the paper. When a source is among the channels, the framework reduces to the relative model, making its advantages questionable.
- The term intractable is misused to describe the likelihood. The issue is not computational hardness but rather the inability to evaluate the likelihood due to the unspecified array geometry. This misuse is further misleading, as intermediate likelihoods are generally intractable; see Equation (7) in [1] and the subsequent explanation.

---

.. [1] Chung, Hyungjin, et al. "Diffusion posterior sampling for general noisy inverse problems." arXiv preprint arXiv:2209.14687 (2022).

**Essential References Not Discussed:**

The seminal work of [1] where that introduces blind diffusion posterior sampling was not mentioned.

---
.. [1] Chung, Hyungjin, et al. "Parallel diffusion models of operator and image for blind inverse problems. IEEE." CVF Conference on Computer Vision and Pattern Recognition. Vol. 1. No. 2. 2023.

**Experimental Designs Or Analyses:**

The experimental design is sound, except for the limitation to only two speakers.
The method should be tested in a multi-speaker setup to better evaluate its effectiveness in more complex separation tasks.

**Methods And Evaluation Criteria:**

- The evaluation methods are fair, with one exception: the experiments were conducted with only two speakers.
The method should also be tested in a multi-speaker setup to assess its performance in more complex scenarios.
- Additionally, the computational requirements were not evaluated.
Given that the procedure involves likelihood estimation, it is important to assess the computational cost of blind speech separation and contrast it with supervised methods.

**Other Comments Or Suggestions:**

Algorithm 1 Issues:
- Contains two ``return`` statements, which may cause confusion.
- the function ``GET_SCORE`` in Line 20 has incorrect arguments. Specifically, it does not take the denoiser as input according to its definition in Algorithm 2

Other Errors:
- Col-2, Lines 158-164: the $\log$ were forgotten before $p$.
- Equation (6): Incorrect SDE; missing a negative sign.
- Gaussian Definition: Incorrect; should have mean 0 and covariance $\sigma^2 I$
- Equation (13): Should use spatial convolution $*_l$

**Other Strengths And Weaknesses:**

- There is confusion in the notation, particularly for the denoiser (Col-2, Line 211), where "s" is uppercase.
This is misleading since uppercase letters seem to be reserved for frequency-domain quantities, yet here it is used to denote the score.
- Algorithm 1 is overly complex: Lines 10 to 25 essentially implement the Heun sampler from EDM, see Algorithm 2 in [1], with the only difference being how the conditional scores (and thus the denoiser) are computed.
It would be clearer to simply state that the algorithm applies the EDM solver with the denoiser described in Algorithm 2, rather than explicitly rewriting these steps.


---

.. [1] Karras, Tero, et al. "Elucidating the design space of diffusion-based generative models." Advances in neural information processing systems 35 (2022): 26565-26577.

**Questions For Authors:**

I have no further question for the authors.

**Relation To Broader Scientific Literature:**

In Table 2, where the microphone array is unknown, the proposed method shows only marginal improvement over supervised methods.
This contradicts the authors' claim that supervised methods lack flexibility and fail to generalize to unseen geometries.
In fact, the results suggest the opposite: the supervised algorithm TF-GridNet-Spatial performs nearly as well as the proposed approach.

**Theoretical Claims:**

The theoretical claims in the main paper are correct.

---

> ### Author Rebuttal · Authors · 2025-03-30
>
> We thank the reviewer for the detailed and constructive comments.
> > Q: The advantage of using a virtual source model is unclear and questionable.
>
> Yes, we understand this concern now. We had tried to explain this but the explanations are scattered in the paper (lines 147-152 (column 1), line 259 (column 2), and then again in Appendix G). We will consolidate these explanations in one place. Briefly, our main reason to propose a virtual source model is as follows:
>
> -- When FCP computes the relative RIR even with respect to the ground truth (reverberant) signal measured at one channel, FCP fails to reconstruct any other channels' sources perfectly (only 11.3 dB of SI-SDR as shown in Table 3).
>
> -- The reason is the relative RIR filter, as mentioned in Eq. 3 (lines 98-99), involves an inverse filtering, for which the inverse may not exist, and may also make the relative filter non-causal. We discuss this in Appendix G (lines 1143-1144).
>
> -- On the other hand, Table 3 row 1 shows that FCP computes the relative RIRs much better when the source estimate is the original anechoic source.
>
> Hence, we proposed to use the virtual source model, hoping that we can sample the anechoic source, thereby leveraging the effectiveness of FCP. We apologize for the confusion; we will explain this in detail early in the paper.
>
> > Q: The term intractable is misused to describe the likelihood.
>
> We think we understand the issue. In previous literature like DPS[1], $p(y|x_t)$ is usually viewed as intractable because it needs to integrate through $x_0$, or $p(x_0|x_t)$ is intractable because it needs to integrate through all intermediate variables. In our case, we viewed $p(y|x_0)$ as intractable because it needs to integrate through all possible operators on $x_0$. Perhaps this was the source of the confusion. We still think that the intractable term applies to our case, but it needs to be appropriately explained. We will modify the paper accordingly.
>
> >Q: The experiments were conducted with only two speakers. More complex scenarios should be tested.
>
> We ran new experiments with 3 speakers on the spatialized WSJ0-3Mix dataset (reverberant) [2]. Below is our result table on the 3-speaker evaluation.
> We also recorded real-world mixtures for the demo.
> **The demo page https://arraydps.github.io/ArrayDPSDemo/ now contains 3-speaker demo and real-world mixtures demo.** More real-world demo samples can be found in https://github.com/ArrayDPS/ArrayDPSDemo/tree/main/demo_results_realworld.
> | Method               |   SDR     |  SI-SDR   |   PESQ     |   ESTOI     |
> |----------------------|-----------|-----------|------------|-------------|
> | Mixture              |  -3.0     |  -3.3     |  1.50      |  0.371      |
> | IVA (Laplacian)      |   5.9     |   3.8     |  2.10      |  0.574      |
> | IVA (Gaussian)       |   9.6     |   7.6     |  2.46      |  0.701      |
> | SpatialCluster       |   6.9     |   5.6     |  2.10      |  0.625      |
> | UNSSOR               |  -2.8     |  -6.1     |  1.50      |  0.313      |
> | ArrayDPS-A           |  12.8 ± 1.2 | 11.8 ± 1.3 | 3.11 ± 0.16 | 0.791 ± 0.028 |
> | ArrayDPS-A-ML        | **14.0**  | **13.0**  | **3.27**   | **0.807**   |
> | ArrayDPS-A-Max       |  14.3     |  13.4     |  3.31      |  0.816      |
> | TF-GridNet-Spatial   |  10.6     |   9.7     |  2.85      |  0.747      |
>
> For the 3-speaker WSJ0-3Mix dataset [2], we are pleasantly surprised that ArrayDPS-A outperforms supervised TF-GridNet-Spatial by a large margin. This shows ArrayDPS's potential in more complicated scenarios.
>
> >Q: Missing citation of BlindDPS[3].
>
> This definitely should be cited. We were supposed to cite this in line 421, second column, when we discussed the blind inverse problem with DPS, but we mistakenly cited DPS again there. Sincere apologies, and thank you for pointing this out.
>
> >Q: Typos in line 211 (confusion for S), second column, line 158-164, second column (missing log), Eq.6 (minus sign), line 136-137, second column (Gaussian typo), Eq.13  typo, line 20 of algorithm 1 (wrong args in get_score).
>
> Thanks! We will fix all these typos in the next version.
>
> >Q: Algorithm 1 is too complex and has 2 return statements.
>
> Thanks for this suggestion. We will abbreviate algorithm 1 and abstract out the second-order Heun sampler. We will also return both signals in the end.
>
> [1] Chung, Hyungjin, et al. *Diffusion posterior sampling for general noisy inverse problems.* arXiv preprint arXiv:2209.14687 (2022).
>
> [2] Mitsubishi Electric Research Laboratories. *Deep Clustering*. https://www.merl.com/research/highlights/deep-clustering
>
> [3] Chung, Hyungjin, et al. *Parallel Diffusion Models of Operator and Image for Blind Inverse Problems*. Proceedings of the IEEE/CVF Conference on Computer Vision and Pattern Recognition (CVPR), Vol. 1, No. 2, 2023.

---

> > ### Comment · Reviewer_Go2n · 2025-04-02
> >
> > I appreciate the authors’ response, which addresses most of my concerns.
> > However, the issue regarding the **Relation to broader scientific literature** remains unresolved.
> >
> > I am confused about the results in Table 2, where the supervised method TF-GridNet performs almost as well as the unsupervised approach in setups with an unknown microphone array.
> > This contradicts the authors’ claim that supervised methods lack flexibility and fail to generalize to unseen geometries.

---

> > > ### Author Response · Authors · 2025-04-03
> > >
> > > >Q: I appreciate the authors’ response, which addresses most of my concerns
> > >
> > > Thanks for your acknowledgment!
> > >
> > > >Q: I am confused about the results in Table 2, where the supervised method TF-GridNet performs almost as well as the unsupervised approach in setups with an unknown microphone array. This contradicts the authors’ claim that supervised methods lack flexibility and fail to generalize to unseen geometries.
> > >
> > > Sorry for missing this important question. In lines 22-25, column 2, we argue that models trained on **fixed array geometries** are not flexible to other arrays (the number of channels and the geometry might not match). However, in Table 2, the supervised TF-GridNet-Spatial is trained on the 4-channel spatialized WSJ0-2Mix dataset, where each mixture’s array geometry is randomized in the dataset (mentioned in Sec. 4.6 line 413 column 1). Thus, TF-GridNet-Spatial in Table 2 is not trained on fixed microphone array geometries and can generalize to any 4-channel microphone arrays (trainset and testset are from the same domain). We are sorry for the confusion and will emphasize this point in Section 4.6.
> > >
> > > Besides that, one potential problem of this kind of supervised generalization (training on ad-hoc arrays) is that it might be trading performance for generalization. We mention this in line 380, right column to line 404, left column.  Basically, in Table 1, row 5(b) and 5(c), TF-GridNet-SMS is trained on a fixed array on 3-channel SMS-WSJ and performs much better than TF-GridNet-Spatial, which is trained on 3-channel ad-hoc arrays for generalization. In contrast to supervised methods that might have a domain mismatch for training and testing, ArrayDPS has no such problem, as there is no training happening, except for a prior diffusion model.
> > >
> > > We would also like to mention that we updated our response to the reviewer L5PQ about array-agnostic baselines. We train a more recent USES [1] on variable-number-of-channel Spatialized WSJ and show the results below (note that USES is trained to generalize to any microphone arrays):
> > >
> > > | USES       | mics| SDR  | SI-SDR | PESQ | ESTOI |
> > > |------------|------------|------|--------|------|-------|
> > > | Spatialized-WSJ| 2          | 12.4 | 11.7   | 3.29 | 0.839 |
> > > | Spatialized-WSJ| 3          | 12.9 | 12.2   | 3.36 | 0.850 |
> > > | Spatialized-WSJ| 4          | 13.1 | 12.4   | 3.39 | 0.854 |
> > > | SMS-WSJ| 3          | 11.3 | 10.6   | 3.01 | 0.818 |
> > > | SMS-WSJ| 6          | 11.3 | 10.7   | 3.03 | 0.820 |
> > >
> > > Thanks again for your insightful observations!
> > >
> > > [1] W. Zhang, K. Saijo, Z. -Q. Wang, S. Watanabe and Y. Qian, "Toward Universal Speech Enhancement For Diverse Input Conditions," 2023 IEEE Automatic Speech Recognition and Understanding Workshop (ASRU), Taipei, Taiwan, 2023, pp. 1-6, doi: 10.1109/ASRU57964.2023.10389733.

---

### Decision · Program_Chairs · 2025-05-01

**Decision:**

Accept (poster)

**Comment:**

The paper proposes a blind source separation model based on diffusion posterior sampling with an approximated likelihood formulation that models the room acoustics and microphone array geometry. Reviewers unanimously suggested publication, having only raised minor concerns regarding clarity and generalization issues, which in my view do not reduce the value of the paper.